# Bridging the Gap Between Cross-Domain Theory and Practical Application: A Case Study on Molecular Dissolution

**Sihan Wang**[1,2]**, Qing Zhu**[1,3]**, Wenjie Du**[1,2,4†]**, Yang Wang**[1,2,4†]

[1]University of Science and Technology of China, China
[2]Suzhou Institute for Advanced Research, USTC, China
[3]Hefei National Research Center for Physical Sciences at the Microscale
[4] State Key Laboratory of Precision and Intelligent Chemistry, USTC
`sihanwang@mail.ustc.edu.cn`
`{duwenjie, qingzhu, angyan}@ustc.edu.cn`

## Abstract

Artificial intelligence (AI) has played a transformative role in chemical research, greatly facilitating the prediction of small molecule properties, simulation of catalytic processes, and material design. These advances are driven by increases in computing power, open source machine learning frameworks, and extensive chemical datasets. However, a persistent challenge is the limited amount of high-quality real-world data, while models calculated based on large amounts of theoretical data are often costly and difficult to deploy, which hinders the applicability of AI models in practical scenarios. In this study, we enhance the prediction of solute-solvent properties by proposing a novel sample selection method: **C**ore **S**ubset **I**terative **E**xtraction (**CSIE**). CSIE iteratively updates the core sample subset based on information gain to remove redundant samples in theoretical data and optimize the performance of the model on real chemical datasets. Furthermore, we introduce an asymmetric molecular interaction graph neural network (AMGNN) that combines positional information and bidirectional edge connections to simulate real-world chemical reaction scenarios to better capture solute-solvent interactions. Experimental results show that our method can accurately extract the core subset and improve the prediction accuracy. Code is available at: https://CISE-AMGNN.

## 1 Introduction

Artificial intelligence (AI) has emerged as a pivotal tool in advancing chemical research [22, 13, 51, 2]. In recent years, AI has demonstrated its transformative potential across various domains, including the prediction of small molecule properties [65, 8], the simulation of catalytic processes [18, 68], and the design and property prediction of materials such as metal-organic frameworks (MOFs) [11, 53]. This progress has been largely fueled by significant advancements in computational power, the proliferation of open-source machine learning frameworks, and the expansion of chemical datasets [38, 16, 3]. The impact of AI on these areas is evident in the rapid growth of related studies and publications.

Despite rapid advances in AI-driven chemistry, a fundamental bottleneck remains the scarcity of high-quality experimental data. Most progress still relies on computational or simulation-based datasets [22, 25], which, while invaluable for method development, often fail to capture the full complexity and variability of practical systems. Large-scale repositories such as GDB-17 [54] and FDB-17 [64] are generated under idealized rules of chemical stability and synthetic feasibility,

---

† : corresponding author

39th Conference on Neural Information Processing Systems (NeurIPS 2025).

neglecting kinetic effects, environmental perturbations, measurement uncertainty, and procedural variation [5]. To mitigate this gap, recent studies have sought to improve generalization under unseen conditions by increasing model capacity or expanding data coverage. For example, Goh et al. [23] employed weakly supervised training on large unlabeled chemical databases combined with transfer learning, while Wang et al. [66] infused chemical reaction information into representation learning.

However, these strategies typically demand ever larger model architectures and datasets, incurring prohibitive computational and storage costs that impede deployment in resource-limited environments. To address both the burden of large-scale training and the challenge of uneven data distributions, researchers in computer vision and natural language processing have developed an alternative paradigm: core subset selection. This approach seeks to identify a small but highly informative subset of the training data, so as to reduce training time and mitigate class or feature space imbalances without substantially degrading performance [26, 60, 57, 58, 50, 4]. By focusing on the most representative or hard examples, core subset selection methods can both accelerate convergence and ensure that under-represented regions of the data manifold receive sufficient attention during learning.

When naively transferred to molecular chemistry, however, existing core subset selection methods exhibit two major shortcomings: ❶Selection criteria based on loss, uncertainty, or feature embeddings learned earlier by the downstream task models tend to overemphasize rare or difficult samples or conversely overlook subtle but chemically important patterns, thereby skewing the training distribution and degrading performance in under-sampled regions of chemical space [28, 32]; ❷ Many algorithms require repeated evaluation of all or a large fraction of samples after each training epoch to update their selection scores, introducing substantial overhead that scales poorly with dataset size, especially in high-throughput molecular screening campaigns [74]. These limitations motivate the development of a chemistry-specific core subset selection framework. Such a framework should be capable of balancing the molecular feature distribution while also minimizing computational costs.

To address the practical limitations of AI-driven chemistry, we identify two complementary challenges: (i) theoretical datasets are large in scale but often noisy, redundant, and imbalanced; (ii) experimental datasets, while high in quality and critical for real-world applications, are limited in size and coverage. These challenges jointly motivate a two-stage solution, illustrated through the case study of dissolution free energy prediction.

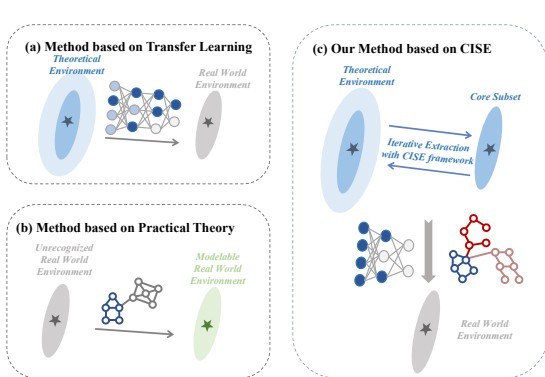

Figure 1: Comparison of methods to assist in bridging the gap between theoretical and practical environments.

In the first stage, we develop Core Subset Iterative Extraction (CSIE), a principled sample selection framework designed to reduce redundancy and improve distributional balance in theoretical datasets. Within the submodular optimization framework [21], CSIE quantifies the value of each sample via information gain, measuring its marginal reduction of uncertainty or entropy [58]. To mitigate the noise and bias inherent in one-shot greedy strategies [20, 27], CSIE adopts an EM-inspired iterative refinement process [40], dynamically re-evaluating sample contributions and progressively approaching a globally optimal subset. For reliable estimation, solute and solvent molecules are independently encoded with a chemical language model, and their embeddings are combined to represent molecular pairs, ensuring efficiency without invoking downstream task models.

Building on the CSIE-selected core subset, we then fine-tune with scarce but reliable experimental data using the Asymmetric Molecular Graph Neural Network (AMGNN). Unlike conventional GNNs, AMGNN explicitly models the directional nature of solute–solvent interactions through distinct bidirectional edges and tailored message-passing schemes. Extensive experiments demonstrate that the integrated CSIE+AMGNN approach significantly improves dissolution free energy prediction and generalizes well to other physical chemistry tasks, such as material band-gap prediction in Appendix E.5.1.

Our contributions can be summarized as follows:

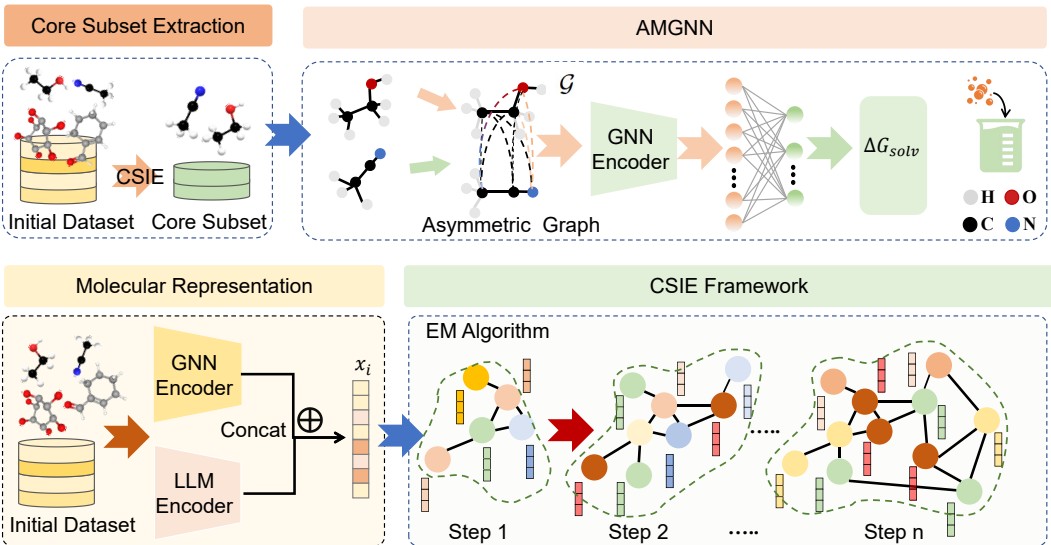

Figure 2: Model framework: For a given solute-solvent dataset, the CSIE framework first extracts the core subset of the dataset (warm points with larger information gain belong to the core subset), and then AMGNN is trained on the core subset to predict the $\Delta G_{\text{solv}}$.

- **CSIE Framework:** We propose an iterative submodular-based core subset selection method that maximizes information gain to reduce redundancy and deviation in theoretical datasets.
- **AMGNN Architecture:** We introduce an asymmetric molecular graph neural network to better simulate directional solute–solvent interactions under realistic chemical conditions.
- **Comprehensive Validation:** We conduct extensive experiments and visualization analyses to demonstrate the performance and interpretability of our method chemical property prediction.

## 2 Method

In this section, we introduce the CSIE framework and the AMGNN model: For a given solute-solvent molecule dataset, in Section 2.1, we use the large model to obtain a multimodal representation of the sample data, thereby providing a basis for CSIE to calculate information gain. In Section 2.2, we describe in detail the process of extracting the core subset of the theoretical dataset using the CSIE framework. Finally, in Section 2.3, we explain how AMGNN simulates real-world solute-solvent interactions to predict $\Delta G_{\text{solv}}$ of the solubility reaction.

### 2.1 Construction of Molecular Representation

Given that many solute and solvent molecules in the dataset share similar SMILES representations yet exhibit significant differences in molecular properties, structural similarity alone is insufficient for accurate sample differentiation. For example, the SMILES of acetic acid is `CC(=O)O`, while that of ethanol is `CCO`. Although they differ by only one oxygen atom in their 2D molecular graphs, their behaviors as solvents are markedly distinct.

To address this, we incorporate the multimodal molecular representation model MolCA [34], which integrates topological and chemical semantic information to generate more expressive embeddings. Molecular text descriptions are primarily sourced from PubChem [30], with additional augmentation from GPT-4 Omni (GPT-4o) [1]. By capturing subtle distinctions between structurally similar molecules, multimodal embeddings substantially improve the accuracy of information gain estimation in the CSIE core subset selection process.

As shown in Figure 2, given an input set composed of the smiles and text description of the solute-solvent pairs $D = \{(g_{solvent1}, t_{solvent1}, g_{solute1}, t_{solute1}), \cdots\}$, it will be encoded by the MolCA:

$$
\begin{aligned}
x_{solute_i} &= \phi_g(g_{solute_i}) \parallel \phi_t(t_{solutei}), \\
x_{solvent_i} &= \phi_g(g_{solvent_i}) \parallel \phi_t(t_{solventi}),
\end{aligned}
\tag{1}
$$

Here, $\|$ represents the concatenation operation of vectors, $g_{solute_i}$ is a preliminary graph vector representation obtained from the smiles of $solute_i$ and $T_{solute_i}$ is a preliminary word vector representation encoded from the textual description of $solute_i$, $\phi_g(\cdot)$ and $\phi_t(\cdot)$ are respectively the graph encoder network and text encoder network of MolCA.

Then we perform a concatenation operation to obtain the embedded representation of the solute-solvent pair $x_i$ by concat the two vectors in Equation (1).

## 2.2 CSIE Framework

Samples with high information gain often correspond to examples near the decision boundary, defined as $m(x) = y \cdot (w \cdot x)$. According to Sorscher et al. [58], when the expected information gain $E[G(x)]$ of the retained samples exceeds the minimum density threshold $c = \inf E[G(x)]$, the generalization error decreases exponentially, following $\epsilon(C_s) \propto e^{-\eta C_s}$, where $C_s$ is the core subset size and $\eta$ is a constant. Furthermore, the resource cost grows only logarithmically with respect to the desired error, i.e., $\propto \ln(1/\epsilon)$, enabling efficient error reduction with limited computational overhead.

In CSIE, we estimate the information gain of each sample as the similarity measure between its embedding representation and surrounding samples, and use this to determine whether it should be retained in the dataset. To efficiently compute similarities among samples and facilitate gain estimation, we employ the Hierarchical Navigable Small World (HNSW) algorithm [36], which allows for scalable and approximate nearest neighbor search in high-dimensional spaces.

For the hierarchical index graph, we use the number of nodes in the insertion layer as the number of latent categories: $C_j = ||l_j|| + 1$. Latent classes can be viewed as classifications of nodes at different levels of topology. Assume that when a node $x_i$ is inserted into the graph, its level $L_j$ is determined by a random process. The node insertion probability $p_j(\mathbf{x}_i)$ of each level $L_j$ can be simulated using a parameterized Bernoulli process:

$$P(\mathbf{x}_i \in L_j) = \frac{e^{-\beta_j}}{Z_j}, Z_j = \sum_{k=0}^{C_j} e^{-\beta_k}. \tag{2}$$

Among them, $\beta_j$ is the attenuation factor related to the level $L_j$, and the normalization factor $Z_j$ is determined by the expected value.

Information gain can be modeled from the perspective of information entropy. First, define the prior information entropy $H(x_i)$ of node $x_i$ before insertion and the posterior information entropy $H(x_i|\mathbf{G})$ after insertion, where $\mathbf{G}$ is the graph structure formed during the insertion process. Information gain $G(x_i)$ can be measured by the difference between the two:

$$G(\mathbf{x}_i) = H(\mathbf{x}_i) - H(\mathbf{x}_i|\mathbf{G}), H(\mathbf{x}_i|\mathbf{Z}_i) = -\sum_{j=0}^{\ell_i} P(H_{ij}) \log P(H_{ij}) \tag{3}$$

where $P(H_{ij})$ is the probability distribution of the implicit category of node $x_i$ at level $L_j$, and $H_{ij}$ represents the category label of node $x_i$ at level $L_j$. Here, we can use the average cosine distance from $x_i$ to its neighboring nodes as an explicit information gain measure. The specific proof can be found in the Appendix B.

From the Equation (3), it can be inferred that these pairs tend to have similar feature encoding to their surrounding pairs. Thus, these paris are often not worth adding to the dataset.

Consider the sample set $\mathcal{X} = \{x_1, x_2, \ldots, x_N\} \subset \mathbb{R}^d$, where each sample $x_i$ follows an unknown distribution $P(x)$. Our goal is to select the core subset $\mathcal{C}_s$ from $\mathcal{X}$ such that its size satisfies:

$$|\mathcal{C}_s| = \lceil s \cdot N \rceil, \quad s \in (0, 1] \tag{4}$$

The core subset $\mathcal{C}_s$ contains the samples with the top $s\%$ information gain, aiming to maximize the contribution to the overall data structure. Since the randomness of the sample insertion order may introduce noise and bias, we draw the idea of EM algorithm to optimize the selection of the core subset, aiming to eliminate the deviation of the insertion order. The two-step iteration steps are as follows:

In the **E-step** of the $t$-th iteration, we select the first $k\%$ high information gain samples from the current core subset $\mathcal{C}_s^{(t)}$ as the unbiased core subset $\mathcal{C}_u^{(t)}$:

$$\mathcal{C}_u^{(t)} = \underset{\substack{\mathcal{C} \subseteq \mathcal{C}_s^{(t)} \\ |\mathcal{C}| = \lceil k \cdot s \cdot N \rceil}}{\arg\max} \sum_{x_i \in \mathcal{C}} G(x_i) \tag{5}$$

In the **M-step**, we first insert $\mathcal{C}_u^{(t)}$ into the HNSW structure, and then randomly insert the remaining sample set $\mathcal{X} \setminus \mathcal{C}_u^{(t)}$, and recalculate the information gain of all samples:

$$G^{(t+1)}(x_i) = \left( \frac{1}{|\mathcal{N}^{(t+1)}(x_i)|} \sum_{x_j \in \mathcal{N}^{(t+1)}(x_i)} dist(x_i, x_j) + \varepsilon \right) \tag{6}$$

The new core subset is updated by maximizing the information gain:

$$\mathcal{C}_s^{(t+1)} = \underset{\substack{\mathcal{C} \subseteq \mathcal{X} \\ |\mathcal{C}| = \lceil s \cdot N \rceil}}{\arg\max} \sum_{x_i \in \mathcal{C}} G^{(t+1)}(x_i) \tag{7}$$

The algorithm ensures the monotonically non-decreasing information gain in each iteration:

$$\sum_{x_i \in \mathcal{C}_s^{(t+1)}} G^{(t+1)}(x_i) \geq \sum_{x_i \in \mathcal{C}_s^{(t)}} G^{(t)}(x_i) \tag{8}$$

The iteration converges when the core subset satisfies the following conditions:

$$\left\| \mathcal{C}_s^{(t+1)} - \mathcal{C}_s^{(t)} \right\|_0 \leq \delta \tag{9}$$

Among them, $\delta$ is the preset tolerance threshold, and $\| \cdot \|_0$ represents the change in the number of collection elements. The proof of the iteration is shown in the Appendix C. The core subset ratio $k$ and EM algorithm threshold $\delta$ are given in the Appendix E.1.

Through this design, our method exhibits properties similar to submodular function ($X \subseteq Y \Rightarrow f(x, X) \geq f(x, Y)$) [21]. Here, $f$ is a set function mapping from the power set of set $\Omega$ to the real number set $R$. $X$ and $Y$ are sets that belongs to $\Omega$ and $x$ satisfies $x \in \Omega \setminus Y$. In this way, proved in Appendix D, Our method satisfies the marginal diminishing effect of the submodular function, which means that it can select the sample set with the highest gain. Due to the characteristics of the HNSW algorithm, it can screen and update samples with a time complexity of $O(log(n))$.

## 2.3 The AMGNN to Predict $\Delta G_{\text{solv}}$

After sample selection by CSIE method, we proposed an asymmetric merged graph and message passing graph neural network to predict $\Delta G_{\text{solv}}$.

We noticed that previous methods often learn chemical reaction behaviors based on solute solvent molecular graphs or merge graphs, which makes it difficult to distinguish between the two situations where molecule $a$ dissolves in molecule $b$ and molecule $b$ dissolves in molecule $a$, because the merge graphs constructed by the two are exactly the same. Therefore, we made a modification: when constructing the merge graph, we chose to add bidirectional directed edges with learnable weights between the topological nodes of the solute and solvent. In this way, during the message passing process, the model can distinguish between the message passing from solute to solvent and from solvent to solute. So that the situations we mentioned above can be distinguished. Specifically, our model AMGNN is as follows:

We use an asymmetric merged graph $\mathcal{G}$ to represent the solvent topology and solute topology connected by directed edges:

$$\mathcal{G} = \{\mathcal{R}, \mathcal{E}, \mathcal{V}, \mathcal{U}\}, \tag{10}$$

Here, $\mathcal{E}$ is the undirected edge of the solute molecule and solvent molecule, $\mathcal{V}$ represents all nodes of in the molecule pair, $\mathcal{U}$ is the global vector representation of the molecule pair, and $\mathcal{R}$ is the set of directed edges between solute and solvent molecule :

$$\mathcal{R} = \{(r_k, a_i, b_j)\}_{k=1}^{2 \times N^a \times N^b}, \tag{11}$$

where $i \in \{1, 2, 3, \ldots, N^a\}$, $j \in \{1, 2, 3, \ldots, N^b\}$. $N^a$ and $N^b$ are the total number of atoms in each of the two molecules. As mentioned above, in order to distinguish between solvent and solute molecules in the merged graph, we constructed this new type of edge $r_k$ as directed edges $r_{ij}$ from $a_i$ to $b_j$ and directed edges $r_{ji}$ from $b_j$ to $a_i$.

Next, following the approach of previous research, we will first perform message transmission and feature updates within the molecule:

$$e'_{ij} = e_{ij} + \tau[\texttt{FC}(v_i + v_j) + \texttt{FC}(e_{ij}) + \texttt{FC}(u)], \quad \hat{e}_{ij} = \frac{\sigma(e'_{ij})}{\sum_{j' \in N_i} \sigma(e'_{ij}) + \epsilon},$$

$$v'_i = v_i + \tau[\texttt{FC}(v_i + \sum_{j \in N_j} \hat{e}_{ij}) \odot \texttt{FC}(v_j) + \texttt{FC}(u)], \tag{12}$$

where $v_i$ and $e_{ij}$ denote the node and edge vectors belonging to $\mathcal{V}$ and $\mathcal{E}$, respectively. $\tau(\cdot)$ and $\sigma(\cdot)$ are activation functions, $\texttt{FC}(\cdot)$ represents a fully connected layer, $\odot$ denotes element-wise multiplication, and $\epsilon$ is a fixed constant.

The next step will be the process of intermolecular information transfer. Compared to previous research, when constructing merged graphs, we add a large number of connecting edges between molecules, which may make message passing and node updates too complex and result in over smoothing. To address this issue, we introduced a mask vector $M = \{m_k | m_k \in \{0, 1\}\}_{k=1}^{2 \times N^a \times N^b}$ during node updates to randomly mask out some edges between molecules:

$$r'_{ij} = \beta_1 * \tau[\texttt{FC}(\delta(v'_{ai}, v'_{bj}) * v'_{ai})] + (1 - \beta_1) * r_{ij},$$

$$r'_{ji} = \beta_2 * \tau[\texttt{FC}(\delta(v'_{bj}, v'_{ai}) * v'_{bj})] + (1 - \beta_2) * r_{ji},$$

$$v''_{ai} = (1 - \beta_3) * v'_{ai} + m_{ji} * \beta_3 * \tau[\sum_{j \in N_b} \texttt{FC}(r'_{ji})], \tag{13}$$

$$v''_{bj} = (1 - \beta_4) * v'_{bj} + m_{ij} * \beta_4 * \tau[\sum_{i \in N_a} \texttt{FC}(r'_{ij})],$$

where $\beta_1, \beta_2, \beta_3, \beta_4$ are learnable parameters that control the update rate of information, which makes information differentiated to avoid information decay or overwrite the original information. $v'_{ai}$ represents the atomic feature vector in the solvent molecule updated by Equation (12) in the merged molecular graph, while $v'_{ai}$ represents the atomic feature vector in the solute molecule. $\delta$ is used to measures the similarity between atoms.

Finally, we update the global feature vector $u$:

$$r'_i = \texttt{FC}(\delta(v'_{ai}, v'_{bj})), \quad u' = u + \tau[\texttt{FC}(\frac{1}{N^v} \sum_{i=1}^{N^v} v'_i + \frac{1}{N^e} \sum_{k=1}^{N^e} e'_k + u)], \tag{14}$$

where $N^v$ and $N^e$ are the number of nodes and edges respectively.

Repeat the above update process until obtaining the final merged molecular feature representation $u_f$. Then we use a MLP to predict predict the $\Delta G_{\text{solv}}$:

$$\Delta G_{\text{solv}} = \texttt{MLP}(u_f). \tag{15}$$

And the expression of the loss function is:

$$\mathcal{L} = D\left(\Delta G_{\text{solv}}; Y\right). \tag{16}$$

where $D$ denotes the distance function, $Y$ is the label value given by the dataset.

## 3 Experimental Results

In this section, we conduct extensive experiments to answer the following questions:

- **RQ1:** Can CSIE and AMGNN make up for the difference between theoretical data and real data?
- **RQ2:** Why can CSIE and AMGNN achieve outstanding results with few samples?
- **RQ3:** Can CSIE also enhance model performance beyond cross-dataset generalization?

## 3.1 Experimental Setup

**Datasets.** The experimental dataset utilized in this study is the *CombiSolv-Exp* database, which is compiled by Vermeire and Green [62], and combined the experimental data from multiple sources. And the *FreeSolv* database is published by Mobley and Guthrie [43], the *CompSol* database from Moine et al. [45], *MNSol* from Marenich et al. [37] and remaining dataset is *Abraham* collected by the Abraham group [24]. The theoretical dataset (CombiSolv-QM) comes from Vermeire et al. [62] which is computerd by the commercial software COSMOtherm.

**Baseline.** To fully validate the effectiveness of our proposed CSIE, we conducted experiments on various models. They are D-MPNN [63], SolvBert [70], SMD [41], Explainable GNN [35], GAT [61], GROVER [52], Uni-Mol [76], Gem [19], which use simple combinations of solute or solvent characterizations without intermolecular messaging. CIGIN [47], CGIB [33] and MMGNN [17] take into account intermolecular messaging and interactions.

**Experimental settings.** The model is trained using the Adam optimizer [31] ($1e^{-4} \to 0.5$) via batch size 50. We employ mean squared error (MSE) as our evaluation indicators. The training process was halted if the validation MSE stop reducing in 150 epochs, or the maximum training limit of 500 epochs was reached. It is implemented using PyTorch via Tesla A100 80GB.

**Evaluation Metrics.** Experiment results are presented in following metrics: the mean absolute error (MAE) and root mean squared error (RMSE). We repeated the training eight times. The mean and standard deviation are recorded.

## 3.2 Generalization Performance From Theoretical Dataset To Experimental Dataset (RQ1)

A **top-1%** core subset QM-mini (same size as CombiSolv-Exp, See the Appendix E.5.2 for more experiments) of the theoretical dataset CombiSolv-QM is selected through the CSIE framework. Referring to previous studies, we compare the performance differences of AMGNN and other baselines on the same test set in two situations: (1) directly train in CombiSolv-Exp, (2) train in QM-mini and then transfer to CombiSolv-Exp. In Table 1, we show the results of randomly partitioning CombiSolv-Exp. Table 2 and Table 3 show the impact of CSIE on the generalization of the model under solvent-split and solute-split. Based on these outcomes, we can delineate three key observations:

**Obs.1: CSIE facilitates the migration of theoretical data to experimental data.** As shown in Table 1, the performance of the model pretrained on QM-mini and then transferred to CombiSolv-Exp has been improved to a certain extent and the performance is more stable, even exceeding the results of previous work [62] using all CombiSolv-QM data for training and migration. Taking the MAE indicator as an example, multiple models such as AMGNN have achieved performance improvements of more than 10%. This fully demonstrates the effectiveness of the CSIE framework. Meanwhile, we give the significance analysis results of Table 1 in the Appendix E.1.

Table 1: Performance comparison on the test set between models trained directly on CombiSolv-Exp and those transferred after training on core subset QM-mini from theoretical CombiSolv-QM.

| Methods | CombiSolv-Exp→CombiSolv-Exp | | QM-mini→CombiSolv-Exp | |
|---|---|---|---|---|
| | MAE ($\downarrow$) | RMSE ($\downarrow$) | MAE ($\downarrow$) | RMSE ($\downarrow$) |
| D-MPNN | $0.456_{(0.042)}$ | $0.672_{(0.051)}$ | $0.402_{(0.017)}$ | $0.613_{(0.026)}$ |
| Explainable GNN | $0.221_{(0.040)}$ | $0.404_{(0.054)}$ | $0.193_{(0.016)}$ | $0.384_{(0.035)}$ |
| SolvBERT | $0.382_{(0.023)}$ | $0.472_{(0.041)}$ | $0.341_{(0.011)}$ | $0.427_{(0.008)}$ |
| GAT | $0.970_{(0.031)}$ | $1.210_{(0.101)}$ | $0.853_{(0.016)}$ | $0.960_{(0.053)}$ |
| GROVER | $0.382_{(0.023)}$ | $0.491_{(0.053)}$ | $0.331_{(0.022)}$ | $0.417_{(0.019)}$ |
| SMD | $0.633_{(0.044)}$ | $1.023_{(0.152)}$ | $0.577_{(0.033)}$ | $0.876_{(0.039)}$ |
| Uni-Mol | $0.214_{(0.022)}$ | $0.373_{(0.043)}$ | $0.203_{(0.021)}$ | $0.311_{(0.027)}$ |
| Gem | $0.253_{(0.023)}$ | $0.551_{(0.023)}$ | $0.218_{(0.016)}$ | $0.499_{(0.031)}$ |
| CIGIN | $0.241_{(0.023)}$ | $0.411_{(0.032)}$ | $0.219_{(0.017)}$ | $0.411_{(0.019)}$ |
| CGIB | $0.223_{(0.037)}$ | $0.381_{(0.030)}$ | $0.211_{(0.025)}$ | $0.355_{(0.025)}$ |
| MMGNN | $0.218_{(0.043)}$ | $0.377_{(0.027)}$ | $0.200_{(0.021)}$ | $0.317_{(0.022)}$ |
| AMGNN | $0.214_{(0.019)}$ | $0.367_{(0.015)}$ | $0.191_{(0.011)}$ | $0.299_{(0.010)}$ |

**Obs.2: CSIE improves the generalization performance of the model.** As shown in Table 2 and Table 3, the performance of the model using CSIE-assisted migration did not show a significant decline in both solvent partitioning and solute partitioning, and was significantly better than the model trained directly on CombiSolv-Exp, especially for some molecules with relatively high specific gravity, such as solvent water and Element O.

**Obs.3: Our AMGNN shows the best performance in various settings.** Taking Random-split as an example, the transfer learning performance of AMGNN improves MAE $4.5\%$ and RMSE $5.6\%$ compared with the sota model MMGNN. Similar improvements were achieved under solvent-split and solute-split. This demonstrates the excellent ability of AMGNN to simulate asymmetric dissolution reactions in the real-world systems.

Table 2: Comparison of the experimental results under solvent-split. The evaluation metric is MAE.

| Solvent | CombiSolv-Exp→ CombiSolv-Exp | | | | | QM-mini→ CombiSolv-Exp | | | | | Test-size |
|---|---|---|---|---|---|---|---|---|---|---|---|
| | GAT | Explainable GNN | CGIB | MMGNN | AMGNN | GAT | Explainable GNN | CGIB | MMGNN | AMGNN | |
| Acetone | $0.303_{(0.032)}$ | $0.284_{(0.023)}$ | $0.252_{(0.024)}$ | $0.232_{(0.014)}$ | $0.213_{(0.012)}$ | $0.271_{(0.032)}$ | $0.249_{(0.013)}$ | $0.206_{(0.032)}$ | $0.192_{(0.012)}$ | $0.186_{(0.009)}$ | 100 |
| Acetonitrile | $0.592_{(0.051)}$ | $0.482_{(0.023)}$ | $0.454_{(0.023)}$ | $0.425_{(0.032)}$ | $0.401_{(0.021)}$ | $0.543_{(0.045)}$ | $0.472_{(0.032)}$ | $0.433_{(0.017)}$ | $0.411_{(0.020)}$ | $0.391_{(0.022)}$ | 67 |
| Benzene | $0.233_{(0.013)}$ | $0.241_{(0.024)}$ | $0.202_{(0.014)}$ | $0.201_{(0.011)}$ | $0.181_{(0.009)}$ | $0.221_{(0.016)}$ | $0.225_{(0.012)}$ | $0.192_{(0.010)}$ | $0.188_{(0.010)}$ | $0.151_{(0.013)}$ | 45 |
| DMSO | $1.217_{(0.164)}$ | $0.977_{(0.074)}$ | $0.955_{(0.083)}$ | $0.915_{(0.062)}$ | $0.885_{(0.054)}$ | $1.163_{(0.083)}$ | $0.870_{(0.032)}$ | $0.855_{(0.071)}$ | $0.802_{(0.030)}$ | $0.793_{(0.048)}$ | 60 |
| Ethanol | $0.291_{(0.024)}$ | $0.262_{(0.021)}$ | $0.253_{(0.022)}$ | $0.251_{(0.029)}$ | $0.226_{(0.018)}$ | $0.279_{(0.026)}$ | $0.255_{(0.016)}$ | $0.222_{(0.014)}$ | $0.225_{(0.019)}$ | $0.206_{(0.053)}$ | 143 |
| Octanol | $0.452_{(0.023)}$ | $0.461_{(0.045)}$ | $0.374_{(0.028)}$ | $0.354_{(0.018)}$ | $0.324_{(0.012)}$ | $0.444_{(0.043)}$ | $0.430_{(0.035)}$ | $0.359_{(0.027)}$ | $0.324_{(0.017)}$ | $0.291_{(0.010)}$ | 226 |
| THF | $0.491_{(0.032)}$ | $0.453_{(0.024)}$ | $0.443_{(0.032)}$ | $0.415_{(0.016)}$ | $0.413_{(0.012)}$ | $0.452_{(0.027)}$ | $0.413_{(0.016)}$ | $0.410_{(0.012)}$ | $0.387_{(0.012)}$ | $0.361_{(0.009)}$ | 116 |
| Water | $2.584_{(0.045)}$ | $2.324_{(0.037)}$ | $2.117_{(0.043)}$ | $2.001_{(0.052)}$ | $1.921_{(0.043)}$ | $2.164_{(0.031)}$ | $1.865_{(0.012)}$ | $1.517_{(0.015)}$ | $1.512_{(0.016)}$ | $1.421_{(0.016)}$ | 1153 |
| Hexane | $0.232_{(0.014)}$ | $0.498_{(0.015)}$ | $0.143_{(0.015)}$ | $0.137_{(0.011)}$ | $0.143_{(0.012)}$ | $0.211_{(0.008)}$ | $0.410_{(0.004)}$ | $0.123_{(0.010)}$ | $0.135_{(0.008)}$ | $0.123_{(0.005)}$ | 186 |

Table 3: The experimental results of solute splitting are compared according to the element type. The evaluation metric is MAE.

| Element | CombiSolv-Exp→CombiSolv-Exp | | | | | QM-mini→CombiSolv-Exp | | | | | Test-size |
|---|---|---|---|---|---|---|---|---|---|---|---|
| | GAT | Explainable GNN | CGIB | MMGNN | AMGNN | GAT | Explainable GNN | CGIB | MMGNN | AMGNN | |
| Br | $0.987_{(0.022)}$ | $0.881_{(0.051)}$ | $0.790_{(0.042)}$ | $0.652_{(0.024)}$ | $0.567_{(0.021)}$ | $0.854_{(0.017)}$ | $0.790_{(0.031)}$ | $0.663_{(0.038)}$ | $0.601_{(0.016)}$ | $0.514_{(0.019)}$ | 152 |
| Cl | $2.043_{(0.022)}$ | $1.696_{(0.051)}$ | $1.256_{(0.042)}$ | $1.276_{(0.024)}$ | $1.186_{(0.027)}$ | $1.721_{(0.015)}$ | $1.244_{(0.016)}$ | $1.112_{(0.009)}$ | $1.078_{(0.014)}$ | $0.942_{(0.014)}$ | 1058 |
| F | $1.201_{(0.043)}$ | $1.211_{(0.051)}$ | $1.053_{(0.023)}$ | $0.913_{(0.029)}$ | $0.914_{(0.031)}$ | $1.031_{(0.016)}$ | $1.042_{(0.019)}$ | $1.066_{(0.033)}$ | $0.801_{(0.024)}$ | $0.718_{(0.023)}$ | 320 |
| I | $0.981_{(0.019)}$ | $0.884_{(0.032)}$ | $0.684_{(0.022)}$ | $0.653_{(0.019)}$ | $0.606_{(0.011)}$ | $0.882_{(0.030)}$ | $0.812_{(0.018)}$ | $0.563_{(0.029)}$ | $0.419_{(0.014)}$ | $0.412_{(0.009)}$ | 117 |
| N | $2.120_{(0.119)}$ | $2.155_{(0.063)}$ | $1.774_{(0.054)}$ | $1.512_{(0.039)}$ | $1.366_{(0.047)}$ | $1.870_{(0.052)}$ | $1.697_{(0.030)}$ | $1.105_{(0.021)}$ | $0.999_{(0.033)}$ | $0.912_{(0.025)}$ | 1192 |
| O | $2.932_{(0.098)}$ | $2.946_{(0.073)}$ | $2.251_{(0.066)}$ | $2.153_{(0.024)}$ | $1.860_{(0.024)}$ | $2.115_{(0.057)}$ | $2.057_{(0.066)}$ | $1.351_{(0.038)}$ | $1.491_{(0.016)}$ | $1.251_{(0.012)}$ | 3796 |
| S | $1.053_{(0.041)}$ | $0.965_{(0.025)}$ | $0.819_{(0.028)}$ | $0.795_{(0.037)}$ | $0.768_{(0.031)}$ | $0.822_{(0.023)}$ | $0.741_{(0.019)}$ | $0.699_{(0.028)}$ | $0.573_{(0.017)}$ | $0.562_{(0.015)}$ | 293 |

## 3.3 Evaluation of The CSIE (RQ2)

To further explore the nature of CSIE, we mapped the features of the samples in CombiSolv-Exp and used the UMAP [39] dimensionality reduction tool to divide them according to solvent type. In Figure 3, we intuitively and clearly show the screening effect of CSIE on samples. Table 4 shows the results of the ablation experiment: For a more intuitive and clear presentation, we do not take transfer learning as an example, but directly perform sample selection, training, and verification on the CombiSolv-Exp dataset to demonstrate the effectiveness of the CSIE framework and AMGNN.

**Obs.4: CSIE removes redundancy by screening samples by category.** As shown in Figure. 3 (a), the samples are distinguished by solvent type, and the dark and light tones indicate whether they are in the core subset screened by the CSIE framework. It can be seen that CSIE has achieved the goal of removing redundant samples while ensuring the quality of the dataset by evenly selecting samples of each category.

**Obs.5: CSIE can balance the distribution of the dataset.** In Figure 3 (b) and Figure 3 (c), we show the proportion of each type of sample before and after the core subset selection by CSIE. It can be clearly seen that through CSIE, the proportion of samples such as 'Aromatics' and 'Halogens', which accounted for a relatively small proportion in the original dataset, has been significantly improved in the core subset, while the proportion of 'Intra-molecular H - bonding', which originally had a large number, has decreased by nearly 10%. This further illustrates that the essence of CSIE is to improve the quality of the dataset by balancing the distribution of various samples in the dataset.

**Obs.6: The samples selected by CSIE are more helpful for training, while the asymmetric network can better simulate the dissolution reaction.** As shown in Table 4, under the w/o CSIE setting, the use of random sampling leads to a huge drop in model performance, especially in the case of few samples. The w/o EM algorithm also degrades the quality of the training set. The model performance also suffers slightly under the w/o Asymmetric network setting using undirected edge modeling.

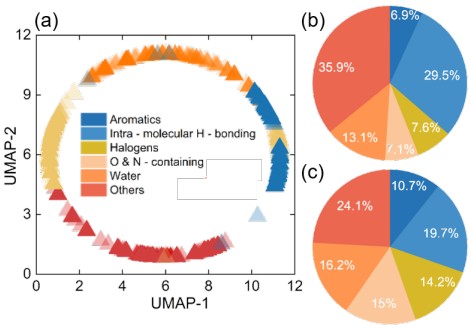

Figure 3: Sample selection and distribution visualization. (a) UMAP projection of solvent molecular embeddings, with lighter points for the full set and darker points for selected samples. (b) Original sample category distribution. (c) Sample category distribution after selection.

Table 4: Ablation experiments are performed on core subsets of different sizes in CombiSolv-Exp to explore the effect of the CISE framework on sample extraction.

| Methods | 20% Trainset | | 40% Trainset | | 60% Trainset | | 80% Trainset | |
|---|---|---|---|---|---|---|---|---|
| | MAE ($\downarrow$) | RMSE ($\downarrow$) | MAE ($\downarrow$) | RMSE ($\downarrow$) | MAE ($\downarrow$) | RMSE ($\downarrow$) | MAE ($\downarrow$) | RMSE ($\downarrow$) |
| w/o CISE | $1.486_{(0.065)}$ | $1.711_{(0.103)}$ | $0.839_{(0.078)}$ | $0.921_{(0.122)}$ | $0.579_{(0.054)}$ | $0.654_{(0.067)}$ | $0.391_{(0.088)}$ | $0.446_{(0.093)}$ |
| w/o EM algorithm | $1.123_{(0.022)}$ | $1.431_{(0.032)}$ | $0.642_{(0.027)}$ | $0.850_{(0.030)}$ | $0.402_{(0.021)}$ | $0.571_{(0.033)}$ | $0.344_{(0.018)}$ | $0.418_{(0.022)}$ |
| w/o Asymmetric network | $0.833_{(0.021)}$ | $0.921_{(0.028)}$ | $0.545_{(0.024)}$ | $0.733_{(0.024)}$ | $0.381_{(0.026)}$ | $0.453_{(0.019)}$ | $0.320_{(0.020)}$ | $0.427_{(0.022)}$ |
| Our Method (CSIE+AMGNN) | $0.777_{(0.011)}$ | $0.811_{(0.009)}$ | $0.311_{(0.020)}$ | $0.350_{(0.018)}$ | $0.247_{(0.015)}$ | $0.261_{(0.015)}$ | $0.313_{(0.017)}$ | $0.370_{(0.027)}$ |

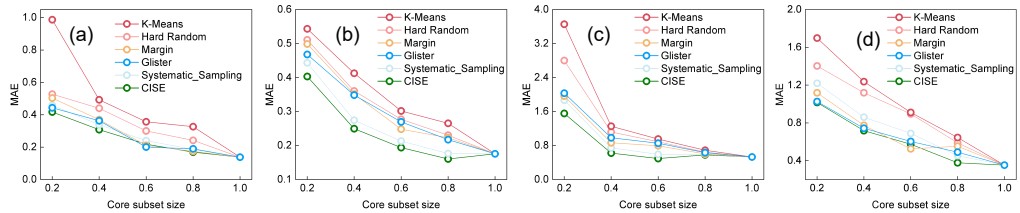

Figure 4: Model performance using various sampling methods under different datasets. **(a)**, **(b)**, **(c)**, and **(d)** represent the experimental results on the Abraham, Compsol, Freesolv, and MNsol datasets.

## 3.4 Performance of CSIE Within The Experimental Dataset (RQ3)

To further explore the universality of the CSIE framework, we trained and tested on four datasets: FreeSolv, MNSol, Abraham, and CompSol. The datasets were randomly divided, and under the premise of keeping the validation set and test set unchanged, we used the CSIE framework and other dataset pruning methods, including Hard Random, Systematic Sampling [69], K-Means [29], Margin [12], Glister [56], to select core subsets of different orders of magnitude for actual training. Figure 4 shows the performance of the model trained on train datasets with different data volumes.

**Obs.7: At the same order of magnitude, the core subset extracted by CSIE is more effective.** From Figure 4 we can see that no matter which dataset or sample size, the model trained with the core subset selected by CSIE outperforms other methods. Our method is far ahead of baseline methods such as random sampling, and only systematic sampling methods are close (this is due to the data arrangement of the dataset itself). This demonstrates the effectiveness of our proposed CSIE in sample selection and its robustness on various datasets.

**Obs.8: CSIE achieves similar performance with fewer samples.** For Abraham (Figure 4 (a)) and MNSol (Figure 4 (d)), the model can achieve performance close to that of using all samples using only 80% of the samples. On the CompSol (Figure 4 (b)) , the performance of training with 80% of the samples even exceeds that of training with all the samples. This is because by iterative core subset selection, CSIE removes some redundant samples that are similar in feature space, effectively avoiding overfitting of the model to a certain type of samples and improving the generalization ability of the model.

**Obs.9: CSIE works well on datasets with a more uniform distribution of categories.** The Freesolv dataset consists of pairs of molecules with different solutes but water as the solvent. Such samples are naturally close in feature space. As can be seen from Figure 4 (c), although the model performance slightly decreases when 80% of the dataset is used, when the training samples are reduced to 60%, the model performance increases instead of decreasing. It indicates that the core subset selected by CSIE effectively reduces the redundant information in the dataset by removing solute molecules with similar properties.

## 4 Conclusion

In the field of physical chemistry, the relationship between theoretical calculations and the real world has always been a difficult problem. This study proposes a core subset iterative extraction (CSIE) framework and an asymmetric molecular interaction graph neural network (AMGNN). CSIE effectively extracts core subsets of theoretical data in a lightweight way, while AMGNN simulates

real molecular reactions through asymmetric connection relationships. Through core subsets and transfer learning techniques, the model can be easily deployed, while expanding the sample space and thus enhancing the model's prediction ability, while balancing computational efficiency, thereby achieving a leap from theoretical data to real experimental results. We verified the effectiveness and interpretability of the proposed method through various experiments.

## 5 Acknowledgement

This paper is partially supported by the National Natural Science Foundation of China (No.12227901). The AI-driven experiments, simulations and model training were performed on the robotic AI-Scientist platform of Chinese Academy of Sciences., Anhui Science Foundation for Distinguished Young Scholars (No.1908085J24), Natural Science Foundation of China (No.62502491).

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

# A Related work

## A.1 Molecular Relational Learning

Molecular relationship learning is typically achieved using graph neural networks (GNNs), with most of them relying on the atomic features within a molecule. For instance, Behler and Parrinello utilized neural networks to model atomic interactions within molecules [6, 7, 10, 9]. Additionally, recent work [49] has combined message-passing and attention mechanisms with GNNs to model molecular representations for predicting solvation free energy. Another framework [33] leverages the graph information bottleneck theory to predict molecular interactions. More sophisticated methods have introduced atom-pair symmetry message-passing functions, which aim to enhance the representation of interatomic interactions and improve the accuracy of molecular behavior predictions [42, 17].

## A.2 Discussion on Graph Neural Network

Graph neural networks (GNNs) [55] have shown strong expressive power in molecular relationship modeling and have been widely used to predict molecular properties, reaction behavior, and solubility free energy. Traditional GNNs model the interactions between atoms and bonds through message passing between nodes to extract the structural and chemical characteristics of molecules. Typical models such as D-MPNN [63] and GAT [61] use this method to complete the representation learning of molecular graphs. Models such as MMGNN [17], CGIB [33] and IGIB [75] try to innovate in modeling the interaction between molecular pairs.

However, in actual chemical systems, there are asymmetric and diverse relationships between molecules (such as solutes and solvents). Traditional GNNs are limited in their ability to model such heterogeneous information. To this end, heterogeneous graph neural networks (HetGNNs) [72] provide an effective solution. HetGNN expands the modeling capabilities of GNN, introduces multiple types of nodes (such as molecules of different categories) and edges (such as hydrogen bonds, hydrophobic interactions, etc.), and supports type-based message passing mechanisms. However, in the field of molecular relationship learning such as dissolution reaction prediction or drug interaction prediction, there is no mature solution for heterogeneous graph construction.

## A.3 From Theoretical Data to Real World

Several approaches have been proposed to bridge the gap between theoretical data and real-world applications. One approach involves training on large-scale theoretical datasets and then using transfer learning techniques to fine-tune the model on smaller experimental datasets [23, 63, 73]. Another strategy focuses on introducing more realistic data processing or modeling methods [71, 14, 15]. For instance, Mohapatra et al. [44] introduced more complex topological structures to capture the similarities among macromolecules, while Wang et al. [66] incorporated chemical reactions into the learning process. Additionally, Stuyver et al. [59] explored the use of quantum mechanics to enhance representations. However, the former requires training on a large amount of theoretical data, which increases the difficulty of model deployment, while the latter increases the complexity of the model and the constraints may not be comprehensive.

# B PROOFS of Information Gain

## B.1 DEFINITION OF INFORMATION GAIN

Information gain quantifies the change in the system's information content before and after the insertion of a node into the graph structure. In the context of HNSW, the insertion of node $\mathbf{x}_i$ affects the distances between the node and its neighboring nodes, which in turn impacts the system's entropy. Before the insertion of node $\mathbf{x}_i$, the distance distribution between each node $\mathbf{x}_j$ and $\mathbf{x}_i$ is denoted by $d(\mathbf{x}_i, \mathbf{x}_j)$, and the entropy is $H(\mathbf{X}_{\text{before}})$. After the insertion, the distances change, and the new entropy is $H(\mathbf{X}_{\text{after}})$. Therefore, the information gain $G(\mathbf{x}_i)$ can be defined as:

$$G(\mathbf{x}_i) = H(\mathbf{X}_{\text{before}}) - H(\mathbf{X}_{\text{after}})$$

This equation shows that the information gain corresponds to the change in entropy before and after the insertion of the node.

## B.2 GRAPH STRUCTURE CHANGES AFTER NODE INSERTION

In the HNSW algorithm, when a node $\mathbf{x}_i$ is inserted, it is connected to several neighboring nodes through the multi-level graph structure. These neighboring nodes influence the overall structure of the graph, particularly the distances between the new node and its neighbors. After the insertion, the distances between all nodes in the graph will change, affecting the overall entropy of the system.

Let $\mathcal{N}(\mathbf{x}_i) = \{\mathbf{y}_1, \mathbf{y}_2, \ldots, \mathbf{y}_m\}$ represent the set of neighboring nodes of node $\mathbf{x}_i$. The distance between node $\mathbf{x}_i$ and its $j$-th neighbor $\mathbf{y}_j$ is given by $d(\mathbf{x}_i, \mathbf{y}_j)$. The average distance between node $\mathbf{x}_i$ and its neighbors is:

$$\bar{d}(\mathbf{x}_i) = \frac{1}{m} \sum_{j=1}^{m} d(\mathbf{x}_i, \mathbf{y}_j) \tag{17}$$

where $d(\mathbf{x}_i, \mathbf{y}_j)$ represents the distance between node $\mathbf{x}_i$ and its $j$-th neighbor $\mathbf{y}_j$.

where $\|\cdot\|$ denotes the Euclidean norm (i.e., the standard vector length). Alternatively, using cosine distance, the distance between nodes $\mathbf{x}_i$ and $\mathbf{y}_j$ is:

$$d_{\cos}(\mathbf{x}_i, \mathbf{y}_j) = 1 - \frac{\mathbf{x}_i \cdot \mathbf{y}_j}{\|\mathbf{x}_i\|\|\mathbf{y}_j\|} \tag{18}$$

where $\cdot$ represents the dot product, and $\|\mathbf{x}_i\|$ and $\|\mathbf{y}_j\|$ are the norms of the vectors $\mathbf{x}_i$ and $\mathbf{y}_j$, respectively.

## B.3 RELATIONSHIP BETWEEN INFORMATION GAIN AND AVERAGE DISTANCE

The insertion of a node influences the average distances between nodes in the graph. Therefore, the information gain is related to the change in the average distance between a node and its neighbors before and after the insertion.

Let $\bar{d}_{\text{before}}(\mathbf{x}_i)$ denote the average distance from node $\mathbf{x}_i$ to its neighbors before insertion, and let $\bar{d}_{\text{after}}(\mathbf{x}_i)$ denote the average distance after insertion. Then, the information gain $G(\mathbf{x}_i)$ can be expressed as:

$$G(\mathbf{x}_i) = \bar{d}_{\text{before}}(\mathbf{x}_i) - \bar{d}_{\text{after}}(\mathbf{x}_i) \tag{19}$$

Thus, the information gain is directly related to the change in the average distance between node $\mathbf{x}_i$ and its neighbors.

## B.4 GEOMETRIC INTERPRETATION OF INFORMATION GAIN

From a geometric perspective, the insertion of node $\mathbf{x}_i$ changes the structure of the graph by modifying the distances between nodes. Information gain quantifies the impact of this change in distance. If, after insertion, the distances between node $\mathbf{x}_i$ and its neighbors decrease (indicating tighter connectivity), the information gain is likely to be higher. Conversely, if the distances increase, the information gain will be smaller.

# C PROOFS of the EM algorithm

## C.1 Proof based on monotonicity

The goal is to find the core subset $\mathcal{C}_s$ of the dataset that accounts for $s\%$ and maximize the function of information gain:

$$L(\mathcal{C}_s) = \log p(\mathcal{C}_u; \mathcal{C}_s) = \log \sum_{\mathcal{C}_u} p(\mathcal{C}_u, \mathcal{C}_s) \tag{20}$$

Introduce the auxiliary distribution $q(\mathcal{C}_u)$ and use Jensen's inequality to construct the lower bound:

$$L(\mathcal{C}_s) = \log \sum_{\mathcal{C}_u} q(\mathcal{C}_u) \frac{p(\mathcal{C}_u, \mathcal{C}_s)}{q(\mathcal{C}_u)} \geq \sum_{\mathcal{C}_u} q(\mathcal{C}_u) \log \frac{p(\mathcal{C}_u, \mathcal{C}_s)}{q(\mathcal{C}_u)} \tag{21}$$

Define the $Q$ function:

$$Q(\mathcal{C}_s^{(t)} | \mathcal{C}_s^{(t-1)}) = \mathbb{E}_{\mathcal{C}_u | \mathcal{C}_s^{(t-1)}} [\log p(\mathcal{C}_u, \mathcal{C}_s^{(t)})] \tag{22}$$

The two steps of the EM algorithm are:

- **E-step:** Take the first $k\%$ samples of the core subset $\mathcal{C}_s^{(t-1)}$ calculated in the previous M steps as the unbiased core subset $\mathcal{C}_u^{(t)}$,

- **M-step:** First insert $\mathcal{C}_u^{(t)}$ into HNSW, then randomly insert the remaining samples and calculate the information gain to determine the new core subset $\mathcal{C}_s^{(t)}$.

Due to the maximization of $Q$ function:

$$Q(\mathcal{C}_s^{(t)}|\mathcal{C}_s^{(t-1)}) \geq Q(\mathcal{C}_s^{(t-1)}|\mathcal{C}_s^{(t-1)}) \tag{23}$$

This results in monotonicity:

$$L(\mathcal{C}_s^{(t)}) \geq L(\mathcal{C}_s^{(t-1)}) \tag{24}$$

Because $L(\mathcal{C}_s)$ has an upper bound, the EM algorithm converges.

### C.2 Proof based on KL divergence

Define the posterior distribution of the unbiased core subset:

$$p(\mathcal{C}_u|\mathcal{C}_s) = \frac{p(\mathcal{C}_u, \mathcal{C}_s)}{p(\mathcal{C}_s)} \tag{25}$$

KL divergence is expressed as:

$$D_{\mathrm{KL}}(q(\mathcal{C}_u) \parallel p(\mathcal{C}_u|\mathcal{C}_s)) = \sum_{\mathcal{C}_u} q(\mathcal{C}_u) \log \frac{q(\mathcal{C}_u)}{p(\mathcal{C}_u|\mathcal{C}_s)} \tag{26}$$

Decompose the objective function:

$$L(\mathcal{C}_s) = Q(\mathcal{C}_s|\mathcal{C}_s^{(t-1)}) + D_{\mathrm{KL}}(q(\mathcal{C}_u) \parallel p(\mathcal{C}_u|\mathcal{C}_s)) \tag{27}$$

**E-step:** Choose $q(\mathcal{C}_u) = p(\mathcal{C}_u|\mathcal{C}_s^{(t-1)})$ to minimize the KL divergence. **M-setp:** Maximize $Q(\mathcal{C}_s|\mathcal{C}_s^{(t-1)})$, thus ensuring:

$$L(\mathcal{C}_s^{(t)}) \geq L(\mathcal{C}_s^{(t-1)}) \tag{28}$$

Since the KL divergence is non-negative, the algorithm converges to a local optimum.

## D  PROOFS of the Submodular Function

In this section, we provide proof details of how data cleaning methods satisfy the representation and properties of submodular function. In section D.1, we elaborate on the representation and properties of submodular function, and define the functional representation of data cleaning method $f_{dc}$. In section D.2, D.3 and D.4, we respectively prove that $f_{dc}$ satisfies three different properties of the submodular function.

### D.1  THE FORM AND PROPERTYS OF THE SUBMODULAR FUNCTION

**The form of the submodular function:** The submodular function is a set function defined on a power set of a finite set $\Omega$. For each given subset, the submodular function returns a real number:

$$f : 2^\Omega \to R, \tag{29}$$

The submodular function should satisfy the following three properties:
**(1) Property 1:** For any $X, Y \subseteq \Omega$, if $X \subseteq Y$, then for all $x \in \Omega \setminus Y$, it should satisfy:

$$f(X \cup \{x\}) - f(X) \geqslant f(Y \cup \{x\}) - f(Y), \tag{30}$$

**(2) Property 2:** For any $S, T \subseteq \Omega$, it should satisfy:

$$f(S) + f(T) \geqslant f(S \cup T) + f(S \cap T), \tag{31}$$

**(3) Property 3:** For any $X \subseteq \Omega, x_1, x_2 \in \Omega$, it should satisfy:

$$f(X \cup \{x_1\}) + f(X \cup \{x_2\}) \geqslant f(X \cup \{x_1, x_2\}) + f(X). \tag{32}$$

Next, based on Equation (E.2) and the form of the submodular function, we further define the mapping method of $f_{dc}$. Given dataset $D$, $f_{dc}$ maps it to the sum of the information gains of all elements in $D$:

$$f_{dc}(D) = G(D_i) = \left( \sum_{x_i \in \mathcal{D}_i} G(x_i) \right), \tag{33}$$

where $\mathcal{N}_x$ includes the $k$-nearest neighbors of $x$ in $D$. $k$ is a fixed parameter given in advance, and the distance between x and its neighboring nodes is calculated based on the cosine similarity of their feature embeddings.

## D.2 DERIVATION PROOF OF THE SUBMODULAR FUNCTION'S PROPERTY 1

According to the definition given in Equation (33), we can conclude that:

$$
\begin{aligned}
& f_{dc}(Y \cup \{x\}) - f_{dc}(Y) \\
=\ & mean(\sum_{y \in \mathcal{N}_x^Y} \frac{x_{mol} \cdot y_{mol}}{||x_{mol}|| \cdot ||y_{mol}||}) \\
=\ & mean(\sum_{p \in \mathcal{N}_x^Y \wedge p \in X} \frac{x_{mol} \cdot p_{mol}}{||x_{mol}|| \cdot ||p_{mol}||} \\
& + \sum_{q \in \mathcal{N}_x^Y \wedge q \in Y \backslash X} \frac{x_{mol} \cdot q_{mol}}{||x_{mol}|| \cdot ||q_{mol}||}) \\
\leqslant\ & mean(\sum_{r \in \mathcal{N}_x^X} \frac{x_{mol} \cdot r_{mol}}{||x_{mol}|| \cdot ||r_{mol}||}) \\
=\ & f(X \cup \{x\}) - f(X).
\end{aligned}
\tag{34}
$$

here, $\mathcal{N}_x^Y$ represents the $k$-nearest neighbors the $k$-nearest neighbors in set $Y$ and $\mathcal{N}_x^X$ represents the $k$-nearest neighbors the $k$-nearest neighbors in set $X$. In the above derivation, the third step is based on a belief that for element $x$, if $Y$ gives new nearest neighbors different from $X$, then it means that the neighbors are elements with a smaller distance from $x$. And only when $\mathcal{N}_x^Y \cap Y = \varnothing$, the third step is to derive an equal sign.

## D.3 DERIVATION PROOF OF THE SUBMODULAR FUNCTION'S PROPERTY 2

Before proving, we first define the set $V$ to represent $S \cap T$. Next, we make the following deduction:

$$
\begin{aligned}
& f_{dc}(S) + f_{dc}(T) \\
=\ & f_{dc}(V \cup S \backslash V) + f_{dc}(V \cup T \backslash V) \\
=\ & f_{dc}(V) + f_{dc}(S \backslash V)_V + f_{dc}(V \cup T \backslash V) \\
\geqslant\ & f_{dc}(V) + f_{dc}(S \backslash V)_{V \cup T \backslash V} + f_{dc}(V \cup T \backslash V) \\
=\ & f_{dc}(V) + f_{dc}(V \cup T \backslash V \cup S \backslash V) \\
=\ & f_{dc}(S \cap T) + f_{dc}(S \cup T).
\end{aligned}
\tag{35}
$$

here, we use the form $f_{dc}(Y)_X$ to represent the sum of information gains that can be obtained by adding elements from $Y$ to the set $X$. That means we can overwrite $f_{dc}(Y)$ as $f_{dc}(Y)_\varnothing$. We provide an explanation for the third step derivation: as we demonstrated in D.2, adding elements to a subset of the set always yields greater benefits than adding elements to the set itself. Therefore, $f_{dc}(S \backslash V)_V \geqslant f_{dc}(S \backslash V)_{V \cup T \backslash V}$ and only when $T \backslash V$ is an empty set, taking an equal sign, which also means $T$ is a subset of $S$ or vice versa.

## D.4 DERIVATION PROOF OF THE SUBMODULAR FUNCTION'S PROPERTY 3

Similar to D.2 and D.3, we prove property 3:

$$
\begin{aligned}
& f(X \cup \{x_1, x_2\}) + f(X) \\
=\ & f(X \cup \{x_1\}) + f(\{x_2\})_{X \cup \{x_1\}} + f(X) \\
\leqslant\ & f(X \cup \{x_1\}) + f(\{x_2\})_X + f(X) \\
=\ & f(X \cup \{x_1\}) + f(X \cup \{x_2\}).
\end{aligned}
\tag{36}
$$

Table 5: Impact of different $k$ values (with $\delta = 5\%$) on information gain and EM convergence.

| $k$ ($\delta = 5\%$) | 0.3 | 0.4 | 0.5 | 0.6 | 0.7 | 0.8 |
|---|---|---|---|---|---|---|
| Final Avg. Information Gain | 0.92 | 0.91 | 0.91 | 0.88 | 0.84 | 0.80 |
| EM Iteration Rounds | 12 | 6 | 3 | 3 | 2 | 1 |

Table 6: Impact of different $\delta$ values (with fixed $k = 0.5$) on information gain and EM convergence.

| $\delta$ ($k = 0.5$) | 5% | 10% | 20% | 30% |
|---|---|---|---|---|
| Final Avg. Information Gain | 0.91 | 0.82 | 0.76 | 0.76 |
| EM Iteration Rounds | 3 | 2 | 1 | 1 |

# E  SUPPLEMENTARY DETAILS

## E.1  HYPERPARAMETERS

This section mainly introduces some hyperparameter settings of the model. For the CISE framework, we set the number of neighbors to 4, the proportion of unbiased core subsets to core subsets k=50%, and the termination condition $\delta$ is 5% of the sum of the information gain of the core subsets in the previous round. For the AMGNN model, we set `GNN_layer_num=3`, `FC_layer_num=3`, `hideen_dim=1600`. The remaining hyperparameters have been given in the experimental settings in the main text.

In addition, we supplemented the hyperparameter experiments of the core subset ratio $k$ and the EM algorithm threshold $\delta$.

CSIE is not highly sensitive to the choice of $k$ according to Table 5, but setting $k$ too low significantly increases the computational cost due to more EM iterations.

We also conducted experiments on the threshold hyperparameter $\delta$ with a fixed $k = 0.5$, as shown in the Table 6. It can be observed that a smaller threshold $\delta$ yields higher information gain, but at the cost of more EM iterations. In contrast, increasing $\delta$ reduces the number of iterations, but may lead to suboptimal convergence. Therefore, a moderate setting (e.g., $\delta = 5\%$ or $10\%$) offers a better trade-off between performance and efficiency.

## E.2  DATASET

In this section, we describe the datasets utilized in this study. The SMILES representations of the molecules are converted into graph structures using the Github code from CIGIN [48]. For datasets associated with solvation free energies, namely MNSol, FreeSolv, CompSol, Abraham, and CombiSolv-Exp, we adopt the SMILES-based datasets provided by previous work [62]. Only solvation free energies measured at a temperature of $298K(\pm2)$ are considered, and ionic liquids and ionic solutes are excluded [62].

- **Abraham** [24] is a dataset compiled by the Abraham research group at University College London. We analyze $6,091$ combinations of $1,038$ solutes and $122$ solvents as per the previous study [62].

- **CompSol** [45] is designed to investigate how hydrogen-bonding interactions influence solvation energies. We consider $3,548$ combinations of $442$ solutes and $259$ solvents from this dataset, following the methodology in [62].

- **MNSol** [37] consists of $3,037$ experimental solvation or transfer free energies of 790 distinct solutes and 92 solvents. For this work, we focus on $2,275$ combinations involving 372 unique solutes and 86 solvents, as done in prior research [62].

- **FreeSolv** [43] offers $643$ experimental and calculated hydration free energies for small molecules in water. This study includes $560$ experimental results based on the dataset from previous work [62].

- **CombiSolv-Exp** [62] includes data from MNSol, FreeSolv, CompSol, and Abraham, amounting to $10,145$ combinations of $1,368$ solutes and $291$ solvents.

- **CombiSolv-QM** [62]. This dataset consists of 1 million randomly selected combinations, incorporating 284 frequently used solvents and 11,029 solutes. The elements present include

Table 7: Atoms (nodes), bonds (edges), and global features for molecular representation

| Atomic features (V) | Bond features (E) | Global features (U) |
|---|---|---|
| Atomic species | Bond type | Total No. of atoms |
| No. of bonds | Conjugated status | Total No. of bonds |
| No. of bonded H atoms | Ring size | Molecular weight |
| Ring status | Stereo-chemistry | – |
| Valence | – | – |
| Aromatic status | – | – |
| Hybridization type | – | – |
| Acceptor status | – | – |
| Donor status | – | – |
| Partial charge | – | – |

Table 8: MAE on the experimental band gap test set (52 GWVD samples) after pretraining on core subsets of the theoretical PBE dataset (546 samples) and subsequent fine-tuning. "Direct" denotes models trained and tested on the experimental data.

| Method | Core subset size (theoretical PBE) | | | | Direct on experimental (no pretraining) |
|---|---|---|---|---|---|
| | 80% | 60% | 40% | 20% | |
| Hard Random | $0.5332_{(0.0308)}$ | $0.5775_{(0.0414)}$ | $0.5759_{(0.0334)}$ | $0.6673_{(0.0876)}$ | — |
| K-means | $0.5511_{(0.0298)}$ | $0.5451_{(0.0457)}$ | $0.6161_{(0.0380)}$ | $0.6457_{(0.0487)}$ | — |
| Glister | $0.5473_{(0.0292)}$ | $0.5234_{(0.0427)}$ | $0.5502_{(0.0388)}$ | $0.5850_{(0.0366)}$ | — |
| Margin | $0.5112_{(0.0305)}$ | $0.5594_{(0.0342)}$ | $0.5675_{(0.0386)}$ | $0.5435_{(0.0201)}$ | — |
| CSIE | $0.4583_{(0.0260)}$ | $0.5002_{(0.0285)}$ | $0.5211_{(0.0264)}$ | $0.4969_{(0.0164)}$ | — |
| w/o Transfer | — | — | — | — | $0.5784_{(0.0618)}$ |

$H, B, C, N, O, F, P, S, Cl, Br$, and $I$. The molar mass of the solutes varies between 2.02 g/mol and 1776.89 g/mol.

### E.3 BASELINE

This section primarily introduces several baseline models involved in the experiment.

**D-MPNN** [62]. This model integrates quantum calculations with experimental data accuracy, utilizing two databases, CombiSolv-QM and CombiSolv-Exp, to predict solvation free energy via transfer learning.

**SolvBert** [70]. It models solute-solvent interactions through combined SMILES representations. Pre-trained on computational solvation free energy datasets, it predicts experimental solvation free energy or solubility by fine-tuning on specific datasets.

**Explainable GNN** [35]. This model incorporates chemically intuitive solvation-related parameters, such as semi-empirical partial atomic charges and solvent dielectric constants, alongside standard atomic and bond-level features. It includes interaction layers that facilitate visualization of solubility-enhancing or -reducing interactions.

**GAT** [61]. A novel neural network architecture designed for graph-structured data, leveraging masked self-attention layers to overcome limitations of traditional graph convolution methods. By allowing nodes to attend to neighboring features, it assigns varying importance to different nodes without expensive matrix operations or prior structural knowledge.

**GROVER** [52]. Extracting rich structural information from extensive unlabeled molecular data, this model utilizes self-supervised tasks with a Transformer-style architecture combined with Message Passing Networks, facilitating efficient training on large-scale datasets to address data scarcity and bias.

**SMD** [41]. Employing quantum charge density of solutes and continuum solvent representation, this approach divides solvation free energy into bulk electrostatic contributions (modeled via IEF-PCM) and short-range interactions within the solvation shell (modeled using atomic surface areas and geometry-dependent constants).

**Uni-Mol** [76]. Featuring two SE(3) Transformer-based pre-trained models, Uni-Mol is trained on 209 million molecular conformations and 3 million protein pocket data. It integrates various fine-tuning strategies for diverse downstream tasks.

Table 9: The effect of transfer learning after obtaining the QM-mini dataset using different data pruning methods, the evaluation metric is MAE.

| Method | QM-mini→CombiSolv-Exp | | | | |
| --- | --- | --- | --- | --- | --- |
| | GAT | Explainable GNN | CGIB | MMGNN | AMGNN |
| K-Means | $1.893_{(0.421)}$ | $0.470_{(0.155)}$ | $0.511_{(0.198)}$ | $0.482_{(0.120)}$ | $0.431_{(0.177)}$ |
| Hard Random | $1.433_{(0.243)}$ | $0.352_{(0.106)}$ | $0.503_{(0.201)}$ | $0.477_{(0.186)}$ | $0.443_{(0.182)}$ |
| Margin | $0.942_{(0.089)}$ | $0.271_{(0.097)}$ | $0.410_{(0.083)}$ | $0.327_{(0.093)}$ | $0.301_{(0.076)}$ |
| Glister | $0.910_{(0.016)}$ | $0.355_{(0.077)}$ | $0.475_{(0.025)}$ | $0.321_{(0.091)}$ | $0.301_{(0.065)}$ |
| Systematic_Sampling | $0.951_{(0.047)}$ | $0.290_{(0.065)}$ | $0.255_{(0.055)}$ | $0.247_{(0.059)}$ | $0.264_{(0.042)}$ |
| CSIE | $0.853_{(0.016)}$ | $0.193_{(0.016)}$ | $0.211_{(0.025)}$ | $0.200_{(0.021)}$ | $0.191_{(0.011)}$ |

Table 10: Significance analysis of the effect of the CSIE framework on different models, with RMSE as the evaluation metric.

| model | GAT | GEM | CGIB | MMGNN | AMGNN |
| --- | --- | --- | --- | --- | --- |
| second-best | $1.210_{(0.101)}$ | $0.551_{(0.023)}$ | $0.381_{(0.030)}$ | $0.377_{(0.027)}$ | $0.367_{(0.015)}$ |
| ours | $0.960_{(0.053)}$ | $0.499_{(0.031)}$ | $0.355_{(0.025)}$ | $0.317_{(0.022)}$ | $0.299_{(0.010)}$ |
| $p$-value | $2 \times 10^{-6}$ | $6 \times 10^{-4}$ | $0.034$ | $1 \times 10^{-4}$ | $2 \times 10^{-7}$ |

**Gem** [19]. Utilizing a geometry-based graph neural network, Gem enhances learning with self-supervised strategies at the geometry level, acquiring comprehensive molecular geometry knowledge for accurate property prediction.

**CIGIN** [47]. This end-to-end framework consists of three phases: message passing, interaction, and prediction, culminating in the final phase to estimate solvation free energies.

**CGIB** [33]. Based on graph conditional information bottleneck theory, CGIB extracts conditional subgraphs to effectively model molecular interactions.

**MMGNN** [17]. Specializing in atomic interactions such as hydrogen bonds, MMGNN initially forms indiscriminate connections between intermolecular atoms, which are refined using an attention-based aggregation method tailored to specific solute-solvent pairs.

### E.4 The Detailed Features For Atoms, Bonds and Molecular Global

Table 7 provides a detailed summary of the chosen atom, bond, and global input features. The process begins with transforming the SMILES strings of both solute and solvent into graph structures using the RDKit library. This library is utilized not only for generating graphs but also for calculating atom and bond features for each graph. To reduce the computational overhead associated with quantum mechanical calculations across the entire dataset, the feature selection was limited to those obtainable via RDKit. To ensure uniformity in the lengths of the bond, atom, and global feature vectors, a linear transformation is applied to each vector prior to initiating the message-passing steps.

### E.5 SUPPLEMENTARY EXPERIMENTS

### E.5.1 EVALUATION OF CSIE ON BANDGAP PREDICTION

**Experimental Setup.** We evaluate CSIE on two complementary band gap datasets from NREL-MatDB [46]. The first, called the *theoretical band gap dataset*, comprises 546 PBE-computed values, and the second, the *experimental band gap dataset*, contains 52 GWVD-computed values.

Table 11: Significance analysis of the effect of the AMGNN and second-best model on different datasets, with RMSE as the evaluation metric.

| dataset | QM-mini | FreeSolv | Abraham | MNsol |
| --- | --- | --- | --- | --- |
| second-best | $0.317_{(0.022)}$ | $0.926_{(0.028)}$ | $0.402_{(0.009)}$ | $0.636_{(0.054)}$ |
| ours | $0.299_{(0.010)}$ | $0.907_{(0.022)}$ | $0.391_{(0.029)}$ | $0.581_{(0.021)}$ |
| $p$-value | $0.032$ | $0.019$ | $1 \times 10^{-4}$ | $0.013$ |

Table 12: Result of different methods on FreeSolv, Compsol, Abraham and MNsol datasets.

| | MAE ($\downarrow$) | | | | RMSE ($\downarrow$) | | | |
|---|---|---|---|---|---|---|---|---|
| | FreeSolv | CompSol | Abraham | MNsol | FreeSolv | CompSol | Abraham | MNsol |
| D-MPNN | $0.710_{(0.057)}$ | $0.198_{(0.015)}$ | $0.499_{(0.040)}$ | $0.938_{(0.036)}$ | $1.276_{(0.061)}$ | $0.370_{(0.019)}$ | $0.687_{(0.026)}$ | $1.422_{(0.084)}$ |
| Explainable GNN | $0.757_{(0.034)}$ | $0.197_{(0.013)}$ | $0.515_{(0.046)}$ | $0.470_{(0.021)}$ | $1.343_{(0.050)}$ | $0.387_{(0.013)}$ | $0.801_{(0.037)}$ | $0.852_{(0.086)}$ |
| SolvBERT | $0.642_{(0.037)}$ | $0.178_{(0.015)}$ | $0.514_{(0.037)}$ | $0.806_{(0.038)}$ | $1.122_{(0.048)}$ | $0.344_{(0.022)}$ | $0.683_{(0.024)}$ | $1.040_{(0.068)}$ |
| GAT | $0.688_{(0.036)}$ | $0.195_{(0.012)}$ | $0.485_{(0.046)}$ | $2.076_{(0.051)}$ | $1.234_{(0.082)}$ | $0.411_{(0.013)}$ | $0.759_{(0.044)}$ | $1.782_{(0.167)}$ |
| GROVER | $0.636_{(0.058)}$ | $0.160_{(0.024)}$ | $0.324_{(0.039)}$ | $0.796_{(0.038)}$ | $1.036_{(0.025)}$ | $0.338_{(0.017)}$ | $0.491_{(0.047)}$ | $0.777_{(0.086)}$ |
| SMD | $0.587_{(0.039)}$ | $0.171_{(0.015)}$ | $0.403_{(0.026)}$ | $1.328_{(0.073)}$ | $1.157_{(0.016)}$ | $0.331_{(0.012)}$ | $0.544_{(0.071)}$ | $1.637_{(0.251)}$ |
| Uni-Mol | $0.577_{(0.041)}$ | $0.167_{(0.029)}$ | $0.343_{(0.078)}$ | $0.436_{(0.036)}$ | $1.023_{(0.071)}$ | $0.314_{(0.021)}$ | $0.629_{(0.039)}$ | $0.792_{(0.070)}$ |
| Gem | $0.606_{(0.045)}$ | $0.186_{(0.012)}$ | $0.222_{(0.072)}$ | $0.522_{(0.038)}$ | $1.172_{(0.065)}$ | $0.303_{(0.020)}$ | $0.681_{(0.033)}$ | $1.162_{(0.038)}$ |
| CIGIN | $0.576_{(0.060)}$ | $0.167_{(0.017)}$ | $0.262_{(0.010)}$ | $0.492_{(0.038)}$ | $0.928_{(0.016)}$ | $0.335_{(0.022)}$ | $0.410_{(0.008)}$ | $0.638_{(0.053)}$ |
| CGIB | $0.563_{(0.035)}$ | $0.164_{(0.015)}$ | $0.211_{(0.006)}$ | $0.434_{(0.053)}$ | $0.957_{(0.024)}$ | $0.291_{(0.020)}$ | $0.411_{(0.007)}$ | $0.648_{(0.051)}$ |
| MMGNN | $0.547_{(0.031)}$ | $0.165_{(0.011)}$ | $0.197_{(0.009)}$ | $0.368_{(0.023)}$ | $0.926_{(0.028)}$ | $0.284_{(0.013)}$ | $0.402_{(0.009)}$ | $0.636_{(0.054)}$ |
| AMGNN | $0.531_{(0.022)}$ | $0.159_{(0.030)}$ | $0.188_{(0.029)}$ | $0.352_{(0.021)}$ | $0.907_{(0.022)}$ | $0.290_{(0.030)}$ | $0.391_{(0.029)}$ | $0.581_{(0.021)}$ |

Table 13: Time complexity of CSIE framework.

| | FreeSolv | CompSol | Abraham | MNsol | CombiSolv-Exp | CombiSolv-QM |
|---|---|---|---|---|---|---|
| **TIME** | 0.1min | 0.1min | 0.1min | 0.05min | 0.4min | 26min |

Each material formula is featurized with `matminer` [67] to extract key descriptors—atomic number, electronegativity, atomic radius, ionization energy, atomic packing efficiency, and valence/conduction band centers—which are then concatenated with MolCA embeddings [34]. After removing entries with missing or invalid values, the theoretical dataset is split 80%/20% for training and validation (seed=42), and performance is measured by MAE.

**Training and Fine-Tuning Protocol.** We apply CSIE to select core subsets of varying sizes from the theoretical dataset, train an MLP (two hidden layers of 100 and 50 ReLU units, 500 epochs) on each subset, and identify the best-performing configuration via validation MAE. This pre-trained model is then fine-tuned on the smaller experimental band gap dataset and finally tested to assess generalization. This two-stage procedure—core set selection on large theoretical data followed by fine-tuning on real-world data—demonstrates CSIE's ability to reduce training cost while preserving predictive accuracy.

**Training Configuration.** We use the Adam optimizer [31] with a learning rate of $1 \times 10^{-3}$ and weight decay of $5 \times 10^{-5}$. The batch size is set to 32, and the model is trained for 100 epochs with early stopping based on test MAE (patience = 20). All experiments are conducted on an NVIDIA A100 GPU. For statistical reliability, each setting is repeated across five random seeds, and we report the mean and standard deviation.

**Baseline Methods.** We compare CSIE against four representative core subset selection methods: Hard Random, K-Means [29], Margin [12] and Glister [56].

**Results and Analysis.1: The core set selection method effectively improves model performance in transfer learning and makes up for the lack of real data.** Table 8 reports the MAE on the experimental GWVD dataset after pretraining on various core-subset sizes of the theoretical PBE dataset and subsequent fine-tuning. Across all subset ratios, CSIE outperforms the direct-training baseline by a substantial margin. In particular, pretraining on 80% of the theoretical data yields MAE $= 0.4583_{(0.0260)}$, an improvement of 0.1201 eV over direct training. Even when using only 20% of the theoretical samples for core subset selection, CSIE achieves MAE $= 0.4969_{(0.0164)}$, still 0.0815 eV better than the direct-training model. The core set selection method represented by CSIE expands the sample space and improves the performance of the model on actual data by effectively screening a large number of theoretical data.

**Results and Analysis.2: The CSIE framework performs best among all core set selection methods.** When compared to other core-subset selection strategies (Hard Random, K-means, Glister, Margin), CSIE consistently delivers lower MAE after transfer across all subset sizes (Table 8). For example, at 40% of the theoretical data, CSIEachieves MAE $= 0.5211_{(0.0264)}$, whereas the next best method Margin yields MAE $= 0.5675_{(0.0386)}$. Moreover, CSIE's standard deviations remain uniformly low, indicating robust performance even under extreme subsampling. These results confirm

Table 14: Training time, memory consumption, and MAE metrics of AMGNN under different settings on the CombiSolv-Exp dataset.

| Metric | AMGNN-100 | AMGNN-80 | AMGNN-60 | AMGNN-40 | AMGNN-20 |
|--------|-----------|----------|----------|----------|----------|
| TIME   | 11.8h     | 8.5h     | 5.7h     | 4.0h     | 3.4h     |
| Memory | 1.6G      | 1.2G     | 1.0G     | 0.9G     | 0.7G     |
| MAE    | 0.214     | 0.216    | 0.220    | 0.234    | 0.266    |

that the information gain-driven selection of CSIE alleviates the problem of uneven distribution of theoretical data, thereby assisting the model to make more accurate and stable predictions under resource-constrained conditions.

### E.5.2   CORE SUBSET RATIO AND TRANSFER IMPACT

The selection of the core subset ratio of 1% of the original data set in the main text is mainly due to two reasons:**(1)** To ensure consistency with the experimental data set scale, while expanding the sample space, the complexity overhead is minimized to the greatest extent; **(2)** The data scale of CombiSolv-QM is relatively large, and a too large core subset will introduce more noise.

In this section, taking AMGNN as an example, the influence of the selection rate s on transfer learning is supplemented. From the results of Table 15 and Figure 4 of the main text, it can be seen that although expanding the core subset ratio increases the number of samples, it may not necessarily improve the performance (it may introduce noise or more samples, resulting in an unbalanced training dataset).

Table 15: Transfer performance and training time cost of AMGNN under different core subset selection ratios $s$ of CombiSolv-QM.

| $s\%$ | 1% | 3% | 5% | 10% | 30% |
|-------|-----|-----|-----|------|------|
| MAE   | $0.191_{(0.011)}$ | $0.194_{(0.008)}$ | $0.186_{(0.010)}$ | $0.194_{(0.011)}$ | $0.200_{(0.005)}$ |
| TIME  | 11.6h | 22.6h | 40.5h | 62.5h | 174.5h |

### E.5.3   SIGNIFICANCE ANALYSIS OF CSIE

The Table 10 shows that the CSIE framework significantly reduces RMSE in all five models, showing consistent performance improvements. Compared with the second-best method, the improvements of CSIE on GAT, GEM, MMGNN, and AMGNN are highly statistically significant ($p < 0.001$), and the improvement on CGIB is also significant ($p = 0.034$). This shows that the CSIE framework has a stable and effective enhancement effect on multiple baseline models.

### E.5.4   SIGNIFICANCE ANALYSIS OF AMGNN

As can be seen from the Table 11, AMGNN outperforms the second-best model on all four datasets, and the RMSE difference is statistically significant. In particular, on the Abraham dataset, the $p$ value is as low as $1 \times 10^{-4}$, indicating that the improvement is extremely significant; on FreeSolv, MNsol, and QM-mini, the $p$ values are also less than $0.05$, indicating that the performance improvement is statistically reliable. This further verifies the robustness and advantages of AMGNN in a variety of molecular property prediction tasks.

### E.5.5   EVALUATION OF DIFFERENT CORESET SELECTION METHODS ON TRANSFER LEARNING

As can be seen from Table 9, different data screening methods have a significant impact on the transfer learning effect. The CSIE method has achieved the lowest MAE on all models, showing the best transfer performance. Compared with traditional methods (such as K-Means and Hard Random), CSIE can significantly reduce the error. For example, on GAT, the MAE of CSIE is 0.853, while K-Means is as high as 1.893, a decrease of more than 50%. In addition, on representative models such

as AMGNN, CSIE is also better than other advanced methods such as Margin and Glister, indicating that this strategy has a stronger ability to retain high-quality information. Overall, CSIE significantly improves the transfer learning effect from QM-mini to CombiSolv-Exp, verifying its robustness and generalization ability in cross-dataset scenarios.

### E.5.6 SUPPLEMENTARY EXPERIMENTS OF AMGNN AND BASELINES ON REAL DATASETS

We run our model and other baseline models on four datasets: FreeSolv, Compsol, Abraham and MNsol in the same experimental setting and training environment as the main text. As can be seen from Table 12, AMGNN outperforms other comparison models on the four datasets, showing the strongest generalization ability and prediction accuracy. Specifically, under the MAE indicator, AMGNN achieved 0.531, 0.159, 0.188 and 0.352 on FreeSolv, CompSol, Abraham and MNsol, respectively, all of which are the best or tied for the best value. Under the RMSE indicator, AMGNN also achieved the best results on Abraham and MNsol, and the other indicators were also very close to the optimal values. This shows that AMGNN can stably adapt to different data distributions and maintain excellent performance in multiple molecular property prediction tasks, verifying its modeling ability and versatility.

### E.5.7 TIME AND SPACE COMPLEXITY AND EFFICIENCY TRADE-OFFS BETWEEN CSIE FRAMEWORK AND AMGNN

In Table 13, we show the time complexity of the CSIE framework. Due to the randomness of the insertion order, We take one iteration of core subset selection as an example, rather than the total time of the final core subset selection. For the efficiency-complexity trade-off of CSIE, see the supplementary hyperparameter experiments and their analysis in the Appendix E.1.

In addition, we use CombiSolv-Exp as an example to supplement the computational efficiency trade-off results when reducing the scale of AMGNN in Table 14. To study the trade-off between computational efficiency and expressive capacity, we evaluate several variants of AMGNN using different inter-molecular edge retention ratios, denoted as AMGNN-20, -40, -60, -80, and -100, corresponding to 20%, 40%, 60%, 80%, and 100% of possible cross-molecular edges being retained. As shown in Table 14, AMGNN-60 achieves a favorable balance, significantly reducing computational cost while preserving prediction accuracy.

### E.5.8 VALIDATION OF AMGNN DESIGN

To validate that AMGNN's improvement stems from its ability to model interaction directionality rather than added model complexity, we compared it against three interpretable baselines. **Flagged GNN**: Adds solute/solvent flags to atom features in a standard GNN. **Heterogeneous GNN**: Uses node types to distinguish solute and solvent atoms. **Embedding Concatenation**: Separately encodes solute and solvent, then concatenates the embeddings for prediction.

Furthermore, we constructed a benchmark subset, **Opposite-Exp**, which includes all 240 solute–solvent pairs in CombiSolv-Exp where both directions exist (i.e., A→B and B→A). This subset directly evaluates model sensitivity to role reversal.

Table 16: Performance comparison of AMGNN and baseline models across multiple datasets.

| Model | FreeSolv | CompSol | Abraham | MNsol | CombiSolv-Exp | Opposite-Exp |
|---|---|---|---|---|---|---|
| Flagged GNN | 0.732 | 0.241 | 0.503 | 0.921 | 0.437 | 0.562 |
| Heterogeneous GNN | 0.581 | 0.183 | 0.258 | 0.439 | 0.254 | 0.294 |
| Embedding Concatenation | 0.785 | 0.197 | 0.462 | 0.453 | 0.223 | 0.367 |
| **AMGNN** | **0.531** | **0.159** | **0.188** | **0.352** | **0.214** | **0.191** |

As shown in Table 16, across all datasets, AMGNN consistently achieves the lowest error, with the most significant improvements observed on the **Opposite-Exp** subset. This confirms that AMGNN effectively captures the asymmetric nature of solute–solvent interactions, a capability not realized by symmetric or type-flagged baselines.

### E.5.9 EFFECTS OF ENCODER ON CSIE

We further investigated how different pre-trained encoders and subset sizes influence the performance of CSIE. Using AMGNN on CombiSolv-Exp as an example, we report results in terms of MAE.

Table 17: Impact of different encoders and subset sizes on CSIE performance on CombiSolv-Exp

| Method | 20% Trainset | 40% Trainset | 60% Trainset | 80% Trainset |
|---|---|---|---|---|
| w/o CSIE (Random) | 1.468 | 0.839 | 0.579 | 0.391 |
| MolCA → GROVER (2D) | 1.032 | 0.667 | 0.457 | 0.353 |
| MolCA → UniMol (3D) | 0.851 | 0.588 | 0.431 | 0.356 |
| MolCA → MolTC (LLM) | 0.789 | 0.321 | 0.365 | 0.324 |
| **Standard CSIE** | **0.777** | **0.311** | **0.247** | **0.320** |

The MAE results of Table 17 indicate that the choice of encoder substantially affects CSIE's effectiveness. In particular, the **LLM-based encoder (MolTC)** outperforms graph-based encoders, and **Standard CSIE achieves the best overall performance**, especially with limited training subsets.

## F    LIMITATIONS

### F.1    DEALING WITH DATA BIAS AND NOISE

The CSIE framework utilizes the hierarchical navigation small world (HNSW) algorithm to compute information gain and select core samples efficiently. However, enhancing its robustness to noisy data remains a key direction. Future improvements could integrate advanced noise detection and removal mechanisms, such as multi-scale clustering or adaptive neighborhood selection, to refine the selection process and enhance training quality.

### F.2    MORE ACCURATE IDENTIFICATION OF OUTLIERS

In the method proposed in this paper, a sample with high information gain means that it is far away from the surrounding samples in space or has low feature similarity. Then, a high-gain sample may be a sample that is added first in a cluster (i.e., a representative sample), but it may also be an outlier. In the future, a more accurate evaluation model should be designed to distinguish the two and further improve the quality of the dataset.

### F.3    EFFICIENT INFORMATION TRANSFER IN AMGNN

The AMGNN model introduces a large number of graph connections to capture solute-solvent interactions, but managing information flow efficiently is an important consideration. Future enhancements could involve adaptive attention mechanisms to selectively prioritize key connections and multi-scale information transfer strategies to balance local and global interactions. These improvements could help mitigate over-smoothing and ensure more effective information propagation, ultimately enhancing solubility prediction accuracy.

## G    Broader Impact

The proposed CSIE-AMGNN framework advances the application of AI in chemistry by improving the efficiency and reliability of molecular property prediction in real-world settings. By introducing a scalable core subset selection method and modeling asymmetric solute-solvent interactions, this work offers a pathway to reduce computational costs while maintaining high predictive accuracy, which is particularly valuable for experimental chemists and material scientists operating under limited data conditions. Potential positive impacts include accelerating the discovery of new solvents, optimizing reaction conditions, and aiding the design of environmentally friendly materials and pharmaceuticals.

