# OpenReview forum: "Bridging the Gap Between Cross-Domain Theory and Practical Application: A Case Study on Molecular Dissolution"
_NeurIPS.cc/2025/Conference — NeurIPS 2025 poster_

### Official Review · Reviewer_vRGh · 2025-06-24

**Clarity:** 2
**Significance:** 3
**Originality:** 3
**Rating:** 5
**Confidence:** 4

**Summary:**

This paper proposes a core subset iterative extraction (CSIE) framework for samDesign of light- and chemically responsive protein assemblies through host-guest interactionsple selection to solve the problem of too large theoretical data sets in transfer learning, and on this basis proposes an asymmetric molecular interaction graph neural network (AMGNN) for predicting solute-solvent properties. Through this method, the article can retain a wider chemical space of theoretical data while reducing the complexity of training. The authors fully verified the effectiveness of the proposed core data set selection method and asymmetric interaction network in two scenarios: solute solvent interaction and band gap prediction.

**Questions:**

- Is it possible that CSIE systematically prefers a certain type of “high-information but more difficult to predict” samples, thereby causing overfitting to these areas during training?

**Ethical Concerns:**

["NO or VERY MINOR ethics concerns only"]

**Limitations:**

yes

**Paper Formatting Concerns:**

There are no particularly obvious problems with the writing format or layout of the article.

**Quality:**

4

**Strengths And Weaknesses:**

**Strengths:**
1. The article has a clear motivation and proposes a novel chemical core subset selection method and molecular interaction network, balancing multiple requirements such as efficiency and overhead, and has good application prospects.

2. The main text and appendix provide sufficient experiments and relatively complete theoretical proofs, and verify the effectiveness of CSIE and AMGNN in multiple scenarios such as solute-solvent reaction and molecular band gap prediction, achieving good results.

3. The article clearly presents the motivation, methods, and results. The content of each section is well-organized and the reading experience is good. The visualization of the methods and conclusions is impressive (Figure 1,3).


**Weaknesses:**
1. The article proposes an asymmetric molecular interaction network, but the discussion on GNN is somewhat thin.

2. Compared with existing subset selection methods based on information theory such as Glister and InfoBatch, the innovation boundary of CSIE is slightly blurred. It is recommended to further emphasize the essential differences and advantages of the method over existing work.

3. Although the author emphasizes advantages such as "lightweight" and "efficient", there is a lack of clear data support, such as how much training time is saved under what scale of data? How does the memory usage change?

---

> ### Author Rebuttal · Authors · 2025-07-29
>
> Thanks for your detailed and constructive review. We sincerely appreciate the time and effort you invested in evaluating our manuscript. We are encouraged by your recognition of the motivation, clarity, and potential impact of our proposed chemical core subset selection strategy and asymmetric molecular interaction network. At the same time, we acknowledge the areas that require further clarification and improvement, particularly in the discussion of GNN-related components, the differentiation from existing information-theoretic methods, and the need for more concrete efficiency metrics. Below, we provide a point-by-point response to your thoughtful comments and suggestions.
>
> ---
>
> > **W1.** Discussion on Graph Neural Network
>
> Thank you for pointing out this issue. We will add the following discussion on GNN in molecular relationship learning:
>
> - Graph neural networks (GNNs) [1] have shown strong expressive power in molecular relationship modeling and have been widely used to predict molecular properties, reaction behavior, and solubility free energy. Traditional GNNs model the interactions between atoms and bonds through message passing between nodes to extract the structural and chemical characteristics of molecules. Typical models such as D-MPNN [2] and GAT [3] use this method to complete the representation learning of molecular graphs. Models such as MMGNN [4] and CGIB [5] try to innovate in modeling the interaction between molecular pairs.
>
> - However, in actual chemical systems, there are asymmetric and diverse relationships between molecules (such as solutes and solvents). Traditional GNNs are limited in their ability to model such heterogeneous information. To this end, heterogeneous graph neural networks (HetGNNs) [6] provide an effective solution. HetGNN expands the modeling capabilities of GNN, introduces multiple types of nodes (such as molecules of different categories) and edges (such as hydrogen bonds, hydrophobic interactions, etc.), and supports type-based message passing mechanisms. However, in the field of molecular relationship learning such as dissolution reaction prediction or drug interaction prediction, there is no mature solution for heterogeneous graph construction.
>
> [1] The graph neural network model.
>
> [2] Transfer learning for solvation free energies: From quantum chemistry to experiments.
>
> [3] Graph attention networks.
>
> [4] MMgnn: A molecular merged graph neural network for explainable solvation free energy prediction.
>
> [5] Conditional graph information bottleneck for molecular relational learning.
>
> [6] Heterogeneous Graph Neural Network.
>
> > **W2.** The difference between CSIE and existing jobs
>
> Although methods such as **Glister** and **InfoBatch** have explored data subset selection strategies based on information gain, gradient diversity or label uncertainty in recent years, these methods still have two fundamental limitations when directly transferred to the field of molecular chemistry, which also constitute the starting point and core innovation of the CSIE framework proposed in this study:
>
> **(1) The weak alignment problem between chemical structure and semantics leads to misleading sample selection by existing methods.** Glister uses the sensitivity estimate of model gradient to generalization error to measure the "importance" of samples, while InfoBatch relies on historical gradient trajectories to dynamically screen samples. However, these methods **are heavily dependent on the representation or loss function feedback learned by the model in the early stages of training**. In chemical molecular data, early representations often fail to capture complex and subtle changes in chemical properties, such as resonance structures, stereo effects, and non-covalent interactions. This leads to: **1. Over-attention to rare or difficult samples** (e.g., pairs of molecules with extremely strong or weak polarity), thereby sacrificing the representativeness of the chemical space; **2. Ignoring important samples that are "fuzzy" in the representation space**, such as molecules with similar topology but significantly different behaviors. The innovation of CSIE is **introducing multimodal semantic embedding (MolCA) as the basic representation**, which is independent of upstream and downstream tasks, and measuring whether the sample brings structural information gain by **information entropy change**, thereby avoiding overfitting dependence on early task models.
>
> **(2) The update computational cost under large-scale data is too high, making it difficult to use for high-throughput chemical screening.**  InfoBatch needs to frequently calculate indicators such as gradient change and loss difference, and Glister needs to evaluate the proxy objective function after each round of training, **which significantly increases the computational overhead in the molecular space (usually tens of thousands of samples**). Especially in practical scenarios, such as drug discovery or solubility screening, training cost is one of the deployment difficulties. The combined strategy proposed by CSIE has the following three efficiency advantages: the approximate nearest neighbor graph (HNSW) in sparse high-dimensional space can efficiently estimate the relative information density between samples; the submodular function structure ensures that the information gain of each round of subset optimization is monotonically constant and the optimization converges; the EM mechanism is introduced to iteratively correct the insertion bias to avoid the local optimum caused by one-time greedy selection. Through these designs, CSIE controls the time complexity to _O(log(n))_, which has the practical potential to process real-scale chemical data.
>
> > **W3.** Time and memory efficiency issues
>
> Regarding the time efficiency, we explain as follows: During the model training process, the CSIE framework reduces the time and space costs mainly by reducing the size of the data set. For example, in the FreeSolv data set, by filtering samples through the CSIE framework, only 60% of the training data is needed to achieve performance that exceeds all training data. Therefore, for this data set, it can be said that CSIE saves 40% of the training cost and achieves better results. In Table 15 of Appendix E.5.2, the superiority of the CSIE framework in transfer learning is more intuitively demonstrated. It can achieve results that exceed 30% of the data with only 5% of the data, and the training time is more than 130 hours. Similarly, smaller data sets will also occupy less memory, but it should be acknowledged that CSIE is a core subset screening framework and does not affect the memory of the model.
>
> ---
>
> > **Q1.** discussion about sample selection of CSIE
>
> It should be clarified that the amount of information is only related to the chemical semantic features of the sample itself, and has nothing to do with the prediction results. This article uses both molecular graph features and semantic features for encoding, and uses the general chemical model MolCA pre-trained on PubChem to fully extract sample features and ensure the completeness of sample chemical semantics as much as possible. Therefore, the CSIE framework does not prefer difficult samples. It prefers to select samples that **are not yet included in the chemical space of the core subset or are in small numbers**. In addition, in Section 3.3, we also visualized the screening effect of CSIE itself on the dataset: we divided the dataset according to solvent type, and evaluated the sample category distribution of the dataset before and after CSIE screening. It can be clearly seen that CSIE can indeed alleviate the redundancy problem of the dataset, obtain a more balanced dataset, and reduce overfitting.
>
> ---
>
> Thank you again for your thoughtful and constructive review. Your feedback has been valuable in helping us improve the clarity, rigor, and positioning of our work. We have carefully revised the manuscript to implement your suggestions. We sincerely hope that our detailed response has adequately addressed your concerns. If the answer is yes, we would be grateful if you would consider revising the overall assessment of the manuscript. Of course, we are happy to provide any further clarification or participate in follow-up discussions as needed.

---

> > ### Comment · Reviewer_vRGh · 2025-08-03
> >
> > Thanks for authors' efforts and detailed response, which has addressed all my concerns – particularly regarding the distinction between CSIE and existing methods. If the authors could include a comparative description of CSIE, GloSTER, and InfoPatch in the revised manuscript, it would help readers better differentiate these approaches. Since the issues have been satisfactorily resolved, I am inclined to accept this paper.

---

> ### Author Response · Authors · 2025-08-04
>
> Dear Reviewer,
>
> Thank you sincerely for your valuable time, thoughtful feedback, and constructive engagement during the review process. Your detailed comments and insightful suggestions have been crucial in enhancing our manuscript's clarity, technical rigor, and overall quality.
>
> Your careful attention to key aspects of our work—such as the motivation and design of the CSIE framework, its relationship with existing GNN-based molecular modeling strategies, and the differentiation from information-theoretic subset selection methods—prompted us to refine our explanations and strengthen both the theoretical and empirical justifications of our approach.
>
> We particularly appreciate your recognition of the novelty and potential impact of the CSIE strategy in chemical data modeling. Your willingness to engage with our point-by-point responses and your encouraging final assessment have motivated our team and reaffirmed the significance of our contribution. As you suggested, we have added a comparative description of CSIE, GloSTER, and InfoPatch in the revised manuscript to help readers clearly understand their differences in methodology, applicability, and efficiency.
>
> Thank you again for your constructive support and guidance. Your feedback has been instrumental in shaping this work, and we truly value your contribution to its development.
>
> Best regards, Authors

---

### Official Review · Reviewer_Lzfx · 2025-06-26

**Clarity:** 1
**Significance:** 1
**Originality:** 1
**Rating:** 3
**Confidence:** 4

**Summary:**

This paper introduces two novel machine learning approaches for modeling solute–solvent interactions: Core Subset Iterative Extraction (CSIE) and the Asymmetric Molecular Graph Neural Network (AMGNN). The authors develop an iterative optimization strategy for selecting a core subset of data points, named CSIE. This method aims to maximize information gain while minimizing redundancy and theoretical bias in the dataset, leading to more representative and efficient training data. To more accurately capture the directional and asymmetric nature of solute–solvent interactions in realistic chemical environments, the authors design a graph neural network AMGNN that incorporates molecular asymmetry into its architecture. Then the authors evaluated CSIE+AMGNN on several datasets.

**Questions:**

1.	Motivation for Dataset Subsampling in Solvation Free Energy Prediction:
The paper introduces a dataset subsampling method (CSIE) for solvation free energy prediction, but it does not sufficiently justify why subsampling is necessary in this domain. Are there any domain-specific challenges—such as noise, redundancy, or systematic bias in theoretical datasets—that make subset selection particularly valuable? If not, why not demonstrate the method’s general utility by applying it to widely-used benchmark datasets like QM9 or MoleculeNet?
2.	Comparison to Full Dataset Training:
Figure 4 shows that training on the full datasets consistently outperforms all subsampling methods on the Abraham, Compsol, Freesolv, and MNsol datasets. What about on the CombiSolv-QM? This raises a fundamental question: if subsampling does not lead to performance gains, what is the practical benefit of CSIE in this context? This is critical to understanding whether CSIE offers any meaningful advantage.
3.	Baseline Comparison for AMGNN:
The effectiveness of the proposed AMGNN architecture is not fully validated. Specifically, the paper should include a comparison to a simple baseline where a standard GNN is used with additional embeddings or flags to mark solute and solvent atoms/molecules or just use heterogeneous graph neural network. Without this comparison, it is unclear whether AMGNN provides any substantial improvement over simpler, more interpretable alternatives.

**Ethical Concerns:**

["NO or VERY MINOR ethics concerns only"]

**Final Justification:**

After carefully considering the author's rebuttal and subsequent discussion, I have revised my rating from 2 to 3. While my initial concerns focused on the lack of cohesion between CSIE and AMGNN, unclear motivation for the architectural design, and marginal performance gains, the authors have provided substantial clarification and supporting evidence on each of these points. Nevertheless, the cohesion issue still exists.

Regarding AMGNN, the authors clarified that the novelty lies not in architectural complexity but in representing chemically meaningful asymmetry via a directed heterogeneous graph. The added baseline comparisons, especially the Opposite-Exp subset, support the claim that this representation captures interaction directionality better than existing methods.

Although the absolute performance improvements remain modest, the authors provided statistical analyses and detailed ablation studies that demonstrate the robustness and significance of their findings. The explanation of how systematic sampling can inadvertently rely on dataset ordering—and how CSIE is more robust under permutation—was insightful and well-supported with additional analysis.

Overall, while some concerns remain about the practical impact of the performance gains, the conceptual contributions and the thoughtful experimental design make this work a meaningful addition to the literature on molecular machine learning. I appreciate the authors' detailed and scientifically grounded responses and have updated my rating accordingly.

**Limitations:**

yes

**Quality:**

2

**Strengths And Weaknesses:**

Strengths:

The paper presents a novel subset selection method, Core Subset Iterative Extraction (CSIE), which offers a potentially useful strategy for improving data efficiency in molecular modeling tasks.

Weaknesses:

1.	Lack of Cohesion and Justification: The paper lacks a clear and coherent structure, and the connections between its core contributions—CSIE and AMGNN—are not well articulated. It remains unclear why CSIE is appropriate for the solvation free energy prediction task, and how it integrates with or enhances AMGNN. As presented, the work feels like two loosely connected efforts rather than a unified framework.

2.	Unclear Motivation for AMGNN Design: The rationale behind the proposed Asymmetric Molecular Graph Neural Network (AMGNN) is insufficiently explained. The paper does not convincingly argue why existing approaches, such as heterogeneous GNNs or simpler methods involving solute–solvent embeddings or interaction labels, are inadequate.

3.	Marginal Performance Gains: The reported improvements over baseline models are not compelling. In Table 1, AMGNN shows only slight performance gains compared to UniMol and Explainable GNN. Similarly, Figure 4 suggests that CSIE offers only marginal benefits over systematic sampling. This weakens the empirical justification for the proposed methods.

---

> ### Author Rebuttal · Authors · 2025-07-29
>
> Thanks for your detailed and constructive comments. We appreciate the time and effort you dedicated to reviewing our manuscript. While we recognize that several aspects required clarification and improvement, we are encouraged by your acknowledgment of the potential value of our proposed subset selection strategy. Below, we provide a point-by-point response to your concerns.
>
> ---
>
> > **W1.** The connection between CSIE and AMGNN
>
> - Both **CSIE** and **AMGNN** are designed to address data quality challenges in AI-driven chemistry. Theoretically generated datasets are often large but noisy, with significant class imbalance, while high-quality experimental data are scarce. To better leverage both types of data, we propose two complementary strategies:
>   **(1)** Use **CSIE** to streamline theoretical datasets by reducing redundancy and improving distribution, thereby lowering pretraining costs.
>   **(2)** Use **AMGNN** to better simulate real-world solute–solvent interactions and enhance model performance on experimental datasets.
>
> - It is important to emphasize that **CSIE is intended as a general-purpose framework** for theoretical data screening. As stated in the title, dissolution free energy prediction serves merely as a case study. We also note in **line 82** that **Appendix E.5.1** includes additional validation on the **material band gap prediction task**, further demonstrating the versatility of CSIE.
>   With such a general framework, we are able to select lighter, better-balanced pretraining datasets (Figure 3), which in turn improve the **performance (Table 1)** and **generalization ability (Tables 2 and 3)** of AMGNN.
>
>
> > **W2 & Q3.** Clarification and Validation of AMGNN Design
>
> -  Design Motivation
>
> 	The core of **AMGNN** is not to propose a new GNN architecture, but to introduce a **more chemically faithful graph representation** by constructing an **asymmetric heterogeneous solute–solvent graph** using DGL.
>
> 	Most existing approaches (e.g., Uni-Mol, Explainable GNN, CIGIN) treat the two molecules **independently**, and then model interactions via **concatenation** or **attention**, which results in a **symmetric treatment** of solute and solvent.
>
> 	However, **molecular interactions are inherently directional**. For instance:
> 	  - *Ethanol dissolves in water*: −5.1 kcal/mol
> 	  - *Water dissolves in ethanol*: −6.3 kcal/mol
>
> 	Such asymmetry is chemically meaningful but cannot be captured by symmetric designs. Therefore, AMGNN directly builds a **directed heterogeneous graph**, which encodes the interaction direction. This enables the model to learn directional dependencies from both structural and relational cues.
>
> - Baseline Comparison & Validation
>
> 	To ensure AMGNN’s superiority is not due to model complexity, we implemented and compared against **three strong, interpretable baselines**:
>
> 	1. **Flagged GNN**: Adds solute/solvent flags to atom features in a standard GNN.
> 	2. **Heterogeneous GNN**: Uses node types to distinguish solute and solvent atoms.
> 	3. **Embedding Concatenation**: Separately encodes solute/solvent, then concatenates for prediction.
>
> 	To further evaluate AMGNN’s ability to model directionality, we constructed a new evaluation subset:
>
> 	- **Opposite-Exp**: Contains all **240 solute–solvent pairs in CombiSolv-Exp** where both directions exist (i.e., A→B and B→A).
> 	- This allows direct testing of the model’s **sensitivity to role reversal**.
>
> |                             | FreeSolv  |  CompSol  |  Abraham  |   MNsol   | CombiSolv-Exp | Opposite-Exp |
> | :-------------------------: | :-------: | :-------: | :-------: | :-------: | :-----------: | :----------: |
> |       **Flagged GNN**       |   0.732   |   0.241   |   0.503   |   0.921   |     0.437     |    0.562     |
> |    **Heterogeneous GNN**    |   0.581   |   0.183   |   0.258   |   0.439   |     0.254     |    0.294     |
> | **Embedding Concatenation** |   0.785   |   0.197   |   0.462   |   0.453   |     0.223     |    0.367     |
> |            AMGNN            | **0.531** | **0.159** | **0.188** | **0.352** |   **0.214**   |  **0.191**   |
>
>  As shown in the table, **our AMGNN consistently outperforms (MAE(↓)) all three approaches, suggesting that the proposed asymmetric design more effectively captures solute–solvent interactions.**
>
>
> > **W3.** Performance improvement analysis
>
> - **Performance Analysis of AMGNN**:
>     - In **Table 1**, since *CombiSolv-Exp* contains a relatively large amount of data, the performance differences among models may not appear very pronounced. However, in **Table 11 on page 20 of the Appendix**, we conduct a necessary **significance analysis**, which confirms that the performance improvement of AMGNN is statistically significant.
>
>     - Furthermore, for the comparison between AMGNN and other models, **Table 12 on page 21 of the Appendix** provides additional validation. On smaller datasets, AMGNN demonstrates a more noticeable advantage.
>
> - **Performance Analysis of CSIE**:
>     - In **Table 10 on page 20 of the Appendix**, we also present the **significance analysis for CSIE**.
>     - In **Observation 7 of Section 3.4**, we note that the comparable performance of **CSIE and systematic sampling** stems from the dataset’s inherent solvent-type organization, which naturally balances categories. As shown in **Table 9 of the Appendix**, systematic sampling offers no significant advantage on _CombiSolv-QM_. To confirm this, we shuffled the sample order in Abraham and Compsol datasets and repeated the experiments; results (marked with *) are presented below.
>
> | Dataset  |       Method        | core subset size=20% | core subset size=40% | core subset size=60% | core subset size=80% |
> | :------: | :-----------------: | :------------------: | :------------------: | :------------------: | :------------------: |
> | Abraham  |        CSIE         |        0.4170        |        0.3080        |        0.2185        |        0.1693        |
> |          | Systematic_Sampling |        0.4587        |        0.3388        |        0.2404        |        0.1862        |
> | Abraham* |        CSIE         |        0.4153        |        0.3102        |        0.2019        |        0.1581        |
> |          | Systematic_Sampling |        0.5028        |        0.4210        |        0.2956        |        0.2433        |
> | Compsol  |        CSIE         |        0.4030        |        0.2496        |        0.1939        |        0.1605        |
> |          | Systematic_Sampling |        0.4433        |        0.2745        |        0.2132        |        0.1765        |
> | Compsol* |        CSIE         |        0.4215        |        0.2352        |        0.1877        |        0.1621        |
> |          | Systematic_Sampling |        0.4995        |        0.3142        |        0.3027        |        0.2018        |
>  It can be seen that:
> >**the CSIE framework is robust when the original dataset is perturbed, while the performance of the method adopted by the system is significantly degraded.**
>
> ---
>
> > **Q1.** Motivation for subset sampling
>
> - In **W1**, we have thoroughly explained the motivation for applying the CSIE framework to dissolution free energy prediction, particularly in collaboration with AMGNN to leverage both theoretical and experimental data. We further demonstrated the generalizability of CSIE across various solute–solvent datasets and in the **Material Band Gap Prediction dataset (Appendix E.5.1)**.
>
> - Moreover, in solute–solvent prediction tasks, the _CombiSolv-QM_ dataset exhibits notable **redundancy and category imbalance**, which can hinder model pretraining and transfer learning. As shown in **Section 3.3 (Figure 3b)**, when samples are grouped by solvent type, **aromatic hydrocarbons make up 29.5%** of the dataset, while **halogen-containing solvents account for only 7.6%**, highlighting the skewed distribution.
>
> > **Q2.** Comparison with full data training
>
> Thanks for your careful observation. However, we would like to clarify that for many datasets, the CSIE subset selection framework effectively balances the sample distribution, **achieving better model performance with fewer training samples**:
>
> - **First**, we sincerely apologize for any confusion caused by the unclear design and analysis of Figure 4. In fact, the results shown in Figure 4 indicate that CSIE achieves **better performance than using the full dataset** on *Compsol*, *Freesolv*, and *MNsol*, while using only **80%**, **60%**, and **80%** of the data respectively. We discuss and analyze this phenomenon in **Obs.8 on page 9 of the main text**.
>
> - **Second**, on *CombiSolv-QM*, we tested the effect of different subset ratios on transfer learning performance. A similar trend was observed as described above. The detailed results and analysis can be found in **Appendix Section E.5.2 and Table 15**.
>
>
> ---------------
>
> We greatly appreciate your insightful and helpful comments, as they will undoubtedly help us improve the quality of our article. If our response has successfully addressed your concerns and clarified any ambiguities, we respectfully hope that you consider raising the score. Should you have any further questions or require additional clarification, we would be delighted to engage in further discussion. Once again, we sincerely appreciate your time and effort in reviewing our manuscript. Your feedback has been invaluable in improving our research.

---

> ### Comment · Reviewer_Lzfx · 2025-08-01
> **Reviewer Response**
>
> Thank you for your detailed and thoughtful rebuttal. I appreciate the effort invested in addressing each of the raised concerns. Below, I provide a brief assessment of each point.
>
> W1. Connection between CSIE and AMGNN
>
> While I appreciate the clarification and the broader context provided regarding the interplay between CSIE and AMGNN, I remain somewhat unsatisfied with this aspect. The manuscript could still benefit from a more explicit articulation of how these two components are jointly motivated and integrated from the outset. That said, I acknowledge the authors' efforts to generalize CSIE beyond a single application and their inclusion of a secondary validation task.
>
> W2 & Q3. AMGNN Design and Validation
>
> This part of the response was well-reasoned and convincing. The clarification regarding the asymmetric solute–solvent interaction modeling and the construction of a directed heterogeneous graph was particularly helpful. The additional baselines and the Opposite-Exp subset offer strong support for the efficacy of the design. I am satisfied with this explanation.
>
> W3. Performance Improvement Analysis
>
> While I appreciate the authors' providing statistical significance analyses, I would like to emphasize that statistical significance alone does not always translate into practical relevance. In particular, when absolute improvements are marginal, the real-world benefit may be limited. The explanation that systematic sampling performs worse after shuffling the dataset is interesting but not entirely clear to me. It would be helpful if the authors could elaborate on why shuffling the sample order causes such degradation in performance.
>
> Q1. Motivation for Subset Sampling
>
> The motivation and justification for CSIE are now clearly articulated. I am satisfied with the explanation and appreciate the attention given to redundancy and imbalance in the dataset.
>
> Q2. Comparison with Full Data Training
>
> The clarification on Figure 4 and the explanation of CSIE’s efficacy relative to full dataset training are appreciated. However, I note that the observed improvements, while statistically valid, remain modest in absolute terms. This somewhat limits the practical impact of the proposed approach. Still, the additional discussion and analysis are helpful.
>
> Overall:
> The rebuttal effectively clarifies most of my previous concerns, particularly regarding the AMGNN design and the motivation for subset sampling. While I still have reservations about the integration of CSIE and AMGNN and the practical impact of performance gains, I recognize the thoughtful response and additional analysis provided. I will consider raising my evaluation score based on this revised understanding.

---

> ### Author Response · Authors · 2025-08-01
>
> Thank you for your thoughtful and constructive feedback. We sincerely appreciate your careful consideration of our responses and your recognition of the clarifications and additional analyses provided. Your comments have helped us better articulate key aspects of our work, and we welcome the opportunity for further dialogue.
>
> ---
>
> >**W1.** Integrated Motivation of CSIE & AMGNN
>
> We sincerely thank the reviewer for their continued engagement and constructive feedback. We now better understand that the core concern lies in the need for a more explicitly unified motivation for CSIE and AMGNN from the outset of the manuscript. We have added a new paragraph to the **Introduction**, before listing our contributions, to make the joint motivation crystal-clear:
>
> > *To overcome the practical limitations of AI-driven chemistry, we begin by identifying two fundamental and complementary challenges: (i) theoretical datasets are large-scale but often noisy, redundant, and imbalanced; (ii) experimental datasets, while high in quality and critical for real-world applications, are limited in size and coverage. These challenges jointly motivate a two-stage solution, illustrated through the case study of dissolution free energy prediction: First, we introduce Core Subset Iterative Extraction (CSIE) to systematically reduce redundancy and improve distributional balance in theoretical datasets, enabling more efficient and informative pretraining. Second, we propose the Asymmetric Molecular Graph Neural Network (AMGNN), which captures directional solute–solvent interactions and fine-tunes model performance based on scarce yet reliable experimental data.*
>
> This framing shows that **CSIE and AMGNN form a unified framework**, jointly addressing the gap between theory and experiment rather than standing alone.
>
> >**W2.** Further analysis of the decline in System_Sampling
>
> Thank you for recognizing our explanation and experiments. We now explain in detail why **systematic sampling** suffers when data order is randomized:
>
> - As we previously mentioned, **the effectiveness of systematic sampling in certain chemical datasets is largely due to the inherent ordering of the data**.
> 	- Systematic sampling selects samples at fixed intervals, and its success relies on a key assumption: the original data has a relatively uniform class distribution along its ordering.
> 	- In some datasets (e.g., **Abraham** and **Compsol**, in the context of solvation free energy prediction), experimental data is often compiled in a series, typically **organized by solvent type**. Systematic sampling can exploit this structure — by tuning the sampling interval, it achieves relatively balanced coverage across categories.
>
> - However, for datasets **lacking this kind of ordering structure** (such as the **bandgap prediction dataset** in the appendix or the **shuffled versions of the solvation datasets**), class distribution along the data order becomes random. In such cases, systematic sampling fails to ensure balanced category representation, leading to **significant performance degradation**.
>
> - To validate the influence of data ordering, we additionally introduced a **shuffled version of the Abraham dataset (denoted as Abraham\*)**, and compared the category distributions selected by **CSIE** and **systematic sampling** under a 60% core subset ratio. The results are shown in the table below.
>
>
> |           Dataset            |       Method        |    Halogens    | Intra-molecular H-bonding | O&N-containing |      Water      |   Aromatics    |      Others      | total |
> | :--------------------------: | :-----------------: | :------------: | :-----------------------: | :------------: | :-------------: | :------------: | :--------------: | :---: |
> |   Distrubition of Abraham    |          -          | **469 (7.7%)** |     **2051 (33.7%)**      | **518 (8.5%)** | **768 (12.6%)** | **259 (4.3%)** | **2026 (33.3%)** | 6091  |
> | Distrubition of 60% Abraham  |        CSIE         |  400 (11.0%)   |       1060 (29.0%)        |  436 (11.9%)   |   613 (19.0%)   |   193 (5.3%)   |   953 (26.1%)    | 3655  |
> |                              | Systematic_Sampling |  384 (10.5%)   |       1045 (28.6%)        |  401 (11.0%)   |   559 (15.3%)   |   206 (5.6%)   |   1059 (29.0%)   | 3654  |
> | Distrubition of 60% Abraham* |        CSIE         |  383 (10.5%)   |       1043 (28.5%)        |  450 (12.3%)   |   590 (16.1%)   |   205 (5.6%)   |   986 (27.0%)    | 3657  |
> |                              | Systematic_Sampling | **284 (7.8%)** |     **1245 (34.1%)**      | **301 (8.2%)** | **459 (12.6%)** | **156 (4.3%)** | **1209 (33.1%)** | 3654  |
>
> From the table, we observe that **systematic sampling exhibits significant category bias** on the shuffled Abraham* dataset. The category distribution after sampling is almost identical to that of the original dataset, which confirms our previous point.

---

> > ### Comment · Reviewer_Lzfx · 2025-08-01
> > **Re**
> >
> > It is much clearer to me now. I am raising my rating.

---

> > > ### Author Response · Authors · 2025-08-02
> > >
> > > Dear Reviewer,
> > > ﻿
> > >
> > > We sincerely appreciate your valuable time and insightful comments, which have significantly improved our manuscript. Your thorough review and constructive feedback at every stage of the process have helped us strengthen our work.
> > > ﻿
> > >
> > > From the initial review to subsequent discussions, your careful attention to all aspects of our research—including the connection between CSIE and AMGNN, the design motivation and validation of AMGNN, performance improvement analysis, and the clarification of subset sampling motivation—has encouraged us to refine our explanations, conduct additional analyses, and enhance the consistency of our framework. These improvements have substantially elevated the quality and robustness of our manuscript.
> > > ﻿
> > >
> > > We are particularly grateful that you recognized the potential value of our CSIE framework and engaged in further discussion. Your willingness to consider our responses point by point and ultimately improve the score has been a great encouragement to us, reinforcing our confidence in the importance of this research.
> > > ﻿
> > >
> > > Once again, we deeply appreciate your time and effort in reviewing our work. Your feedback has been instrumental in shaping this manuscript, and we are truly thankful for your support.
> > >
> > >
> > > Best regards, Authors.

---

### Official Review · Reviewer_2Vf7 · 2025-07-02

**Clarity:** 3
**Significance:** 3
**Originality:** 3
**Rating:** 4
**Confidence:** 4

**Summary:**

Addressing the mismatch between large simulated datasets and scarce experimental data, the authors introduce a two-part framework. Core Subset Iterative Extraction iteratively selects a compact, information-rich subset of theoretical solute–solvent data using submodular information-gain criteria computed from pretrained chemical language-model embeddings. The selected subset is then used to train an Asymmetric Molecular GNN that models directional solute→solvent and solvent→solute message passing. Together, CSIE and AMGNN improve experimental-data accuracy and reduce training cost, and the selection strategy generalizes to a second case study on band-gap prediction.

**Questions:**

1 How sensitive is the CSIE to the choice of the pretrained chemical encoder and the size of subset selected? Also, did the authors experiment with different core subset sizes and how does that affect the final model accuracy?

2 Can the authors clarify the individual contributions of CSIE and AMGNN to the performance gain? Providing an ablation where (a) no CSIE + AMGNN, (b) CSIE + standard GNN, and (c) CSIE + AMGNN are compared on the real data would isolate how much each part helps.

3  The method assumes theoretical data has useful information but includes redundancy. What if the theoretical calculations systematically deviate from experimental reality (e.g., consistently overestimate dissolution free energies or fail for certain classes of compounds)? Would the model still be biased?

4.  The approach was tested on dissolution and briefly on band gap. Could the authors provide more detail on the latter or other domains – for example, did CSIE need any modification for selecting theoretical band-gap data? And does AMGNN’s asymmetry concept apply there (or was only CSIE used)?

**Ethical Concerns:**

["NO or VERY MINOR ethics concerns only"]

**Final Justification:**

Thanks authors for informative and careful rebuttal. Most of my concerns has been addressed, I raise my score to 4.

**Limitations:**

Yes

**Quality:**

3

**Strengths And Weaknesses:**

Strengths:
1. The CSIE method is grounded in submodular optimization – it iteratively adds samples that maximize information gain, which is a principled way to reduce redundancy in the simulation data. The second component, AMGNN, is well-motivated: real solute–solvent interactions are asymmetric.

2. The paper’s experiments show that combining CSIE + AMGNN yields improved prediction accuracy on real-world dissolution data, indicating the approach successfully bridges the gap as intended.

Weaknesses:
1. The novelty is split across two techniques, making each part feel somewhat thin in isolation. CSIE draws from known concepts in active learning, and AMGNN is essentially a tailored GNN architecture. The question is if either component by itself is a significant advance.

2. Another weakness is the potential dependence on the quality of the pre-trained chemical encoder used for computing sample diversity – the paper uses a “chemical LLM” to embed molecules for CSIE. If that encoder has biases or limitations (e.g., it might not encode solvation-specific features), CSIE might pick suboptimal subsets. Also, CSIE is evaluated by final model performance, but it would be better to see an intrinsic evaluation: e.g., does the core subset indeed have less redundancy or better cover important solute–solvent feature space than random sampling?

3 The focus is on dissolution; even though they argue it’s a general approach, the paper reads a bit like a case study rather than a broad benchmark.

---

> ### Author Rebuttal · Authors · 2025-07-29
>
> Thanks for your thoughtful and constructive feedback. We sincerely appreciate the time and effort you invested in reviewing our manuscript. We are encouraged by your recognition of the principled basis of our subset selection strategy and its empirical effectiveness in bridging theoretical and experimental domains. At the same time, we acknowledge that some aspects require further clarification. Below we provide a detailed, point-by-point response to your comments and suggestions.
>
> ___
>
> > **W1.&W3.** Novelty and Generality of CSIE and AMGNN
>
> Thank you for the valuable comments. While **CSIE** and **AMGNN** are two distinct components, they are **proposed in tandem to address a core challenge in AI chemistry**: the gap between large, imbalanced theoretical datasets and limited, high-quality experimental data. CSIE focuses on **data quality and efficiency**, while AMGNN targets **accurate modeling of molecular interactions**. Their integration offers a coherent solution for effectively leveraging both theoretical and experimental data.
>
> We emphasize that **both methods have standalone value**:
>
> - **CSIE** is not a typical active learning strategy. It **avoids uncertainty-based sampling**, which is unreliable in noisy chemical domains, and instead adopts a **structure-aware, distribution-optimized selection** mechanism. Its **generality is validated on a separate task**—material band gap prediction (Appendix E.5.1)—demonstrating that it is **not limited to dissolution**.
>
> - **AMGNN** introduces a **novel asymmetric graph structure** to model directional interactions between solute and solvent, addressing a key limitation in prior work that **assumed symmetric or simplistic combinations**. This is **crucial in chemistry**, where interaction order significantly impacts properties (e.g., solvation, drug interactions).
>
> Although **dissolution prediction** is the main benchmark in this paper, it was selected due to the **availability of both theoretical and experimental data**, making it suitable for evaluating both CSIE and AMGNN. However, the **methods are general** and can be **readily extended to other molecular property prediction tasks**, as demonstrated by our additional experiments.
>
> In summary, **CSIE and AMGNN are unified by a shared objective**, each solving a different but complementary part of the same problem. **Their combination is synergistic, but each also contributes meaningful innovation individually**.
>
>
> > **W2.** Sampling quality of CSIE
>
> Thank you for pointing out the problem about CSIE sampling quality.
>
> - First of all, it should be noted that the encoder does affect CSIE's evaluation of samples to a certain extent, which is also a problem we hope to alleviate. For this reason, this paper uses both molecular graph features and semantic features for encoding, and uses the general chemical model MolCA pre-trained on PubChem to alleviate the problem of domain bias.
>
> - Secondly, **in Section 3.3 Fig 3**, we also visualized the screening effect of CSIE itself on the dataset: we evaluated the sample category distribution of the dataset before and after CSIE screening according to the solvent type. It can be clearly seen that CSIE can indeed **alleviate the redundancy problem of the dataset and obtain a more balanced dataset.**
>
> ___
>
> > **Q1.** Effects of encoder and subset size on CSIE
>
> - The following table uses AMGNN and CombiSolv-Exp as examples to supplement the impact of pre-trained encoders on CSIE (indicator is MAE). It can be seen that the encoder will change the effect of the subset selected by CSIE to a certain extent, and LLM is significantly better than GNN encoder.
>
> |          Method          | 20% Trainset | 40% Trainset | 60% Trainset | 80% Trainset |
> | :----------------------: | :----------: | :----------: | :----------: | :----------: |
> |    w/o CSIE (Random)     |    1.468     |    0.839     |    0.579     |    0.391     |
> | MolCA->GROVER(2D graph)  |    1.032     |    0.667     |    0.457     |    0.353     |
> | MolCA->UniMol (3D graph) |    0.851     |    0.588     |    0.431     |    0.356     |
> |    MolCA->MolTC (LLM)    |    0.789     |    0.321     |    0.365     |    0.324     |
> |      Standard CSIE       |    0.777     |    0.311     |    0.247     |    0.320     |
> - Regarding the sizes of different core subsets, we provide the experimental results and corresponding analysis on different data sets and different core subset sizes in Figure 4 in Section 3.4 of the main text and Table 15 in Section E.5.2 of the Appendix. We can draw the following conclusion: The larger the core subset, the better the effect. On multiple data sets, smaller core subsets have achieved better results, such as Compsol (80%), FreeSolv (60%), and CombiSolv-QM (5%).
>
> > **Q2.** Ablation experiment
>
> We provide the corresponding ablation experiments in **Table 4 on page 9** of our paper:
>
> 1. **w/o CSIE** versus **no CSIE+AMGNN**: Instead of using CSIE and performing multiple experiments, we use a randomly selected dataset and an AMGNN network.
>
> 2. **Asymmetric network** versus **CSIE+standard GNN**: We eliminate all interaction edges and use only the basic MPNN mechanism to extract individual molecular information, then obtain the overall bimolecular information through concatenation.
>
> 3.  **Our Method (CSIE+AMGNN)** versus **CSIE+AMGNN**
>
> The comparative results in Table 4 fully demonstrate the respective roles of CSIE and AMGNN. If you would like to add more ablation experiments on other datasets or models to further resolve your confusion, we are also happy to discuss with you and conduct more experimental verification to further improve our paper. Thank you very much for your help!
>
> > **Q3.** The impact of theoretical calculation deviation
>
> Thank you for your concern about this. We will explain from two perspectives that our model is not easily affected by theoretical calculation bias. First, the CSIE framework evaluates information and redundancy based solely on sample characteristics, without considering the impact of theoretical dataset labels. Second, the pre-training task focuses on encoder pre-training, while the task prediction network is carefully fine-tuned on real data.
>
> To verify this, we conducted additional experiments: **We randomly varied the calculated results (label values) on the theoretical CombiSolv-QM dataset from 0 to 10%, 20%, 30%, 40%, 50% and presented the transfer learning results on AMGNN.**
>
> |     | original dataset | ±10%  |   ±20%    | ±30%  | ±40%  | ±50%  |
> | :-: | :--------------: | :---: | :-------: | :---: | :---: | ----- |
> | MAE |      0.191       | 0.193 | **0.192** | 0.201 | 0.196 | 0.213 |
>
> Generally speaking, for the same sample, the error between theoretical calculation and actual experiment is **within ±25%,** and the CSIE framework still maintains stable effectiveness. Of course, as random errors are amplified, the encoder will also be affected to a certain extent, resulting in deviations.
>
> > **Q4.** Explanation of some details
>
> Band gap prediction is a single-molecule task. For data screening, CSIE only needs to modify the final bimolecular feature concatenation step and use only single-molecule features for information gain calculation. Regarding the model architecture, since this is a single-molecule task, we only use a simple GCN+MLP for prediction.
> ___
>
> Thanks again for your time and effort in reviewing our manuscript. Your feedback is important to our efforts to improve our work. If our response adequately addresses your concerns, we sincerely hope that you will consider revising your rating. We are also happy to provide further clarification or continue the discussion.

---

> ### Author Response · Authors · 2025-08-05
> **Looking forward to your comments and further discussion**
>
> Dear Reviewer,
>
> Thanks once again for your careful reading and insightful feedback on our manuscript. We have now submitted a detailed response that we hope addresses your concerns thoroughly.
>
> Specifically, we clarify the independent contributions of CSIE and AMGNN, justify our data and model design choices, and add new empirical results. We hope that these additions will help resolve previous ambiguities.
>
> If there are any remaining questions or suggestions, we would greatly appreciate the opportunity to address them. Otherwise, if you find our response satisfactory, we would be grateful if you could consider updating your review accordingly.
>
> Thank you once more for your valuable comments, which have significantly contributed to improving our manuscript.
>
> **Best regards,**
>
> *The authors*

---

> ### Author Response · Authors · 2025-08-07
>
> Dear Reviewer 2Vf7,
>
> We sincerely appreciate the time and effort you have devoted to reviewing our manuscript. Previously, we submitted a detailed response addressing all the concerns you raised, including additional experiments, clarifications, and details throughout the paper.
>
>  **The discussion will end in about two days, but we have not yet received your feedback regarding whether our responses have adequately resolved your concerns so far.** We would like to kindly follow up to confirm if our replies have sufficiently addressed your points. If there are still any unresolved issues or areas requiring further clarification, please do not hesitate to let us know—we remain fully committed to making additional improvements to ensure the work meets your expectations.
>
> If our replies have satisfactorily resolved your concerns, we would be truly grateful if you could consider reflecting this in your evaluation by providing a positive score. Your feedback is of great significance to us, and your support would mean a great deal.
>
> Thank you once again for your constructive comments and for contributing to the improvement of our work. We sincerely look forward to your response.
>
> Warm regards,
>
> The Authors

---

> ### Comment · Reviewer_2Vf7 · 2025-08-08
>
> Thanks for the rebuttal, I have raised my score to 4.

---

> ### Author Response · Authors · 2025-08-08
>
> Dear Reviewer,
>
> Thank you very much for your constructive feedback and for the time and effort you have invested in reviewing our manuscript. We are greatly encouraged by your recognition of the principled basis of our subset selection strategy and its empirical effectiveness.
>
> In particular, we appreciate your valuable comments on the impact of encoders and theoretical biases in the dataset on the results of the CSIE framework, which are highly instructive for enhancing the quality of our research.
>
> Thank you again for your support and careful guidance. We look forward to the opportunity to improve our work based on your suggestions and sincerely appreciate that you consider updating your rating of the manuscript.
>
> Sincerely,
>
> The Authors

---

### Official Review · Reviewer_24Q7 · 2025-07-03

**Clarity:** 3
**Significance:** 3
**Originality:** 3
**Rating:** 4
**Confidence:** 2

**Summary:**

This paper proposes a new sample selection framework, Core Subset Iterative Extraction (CSIE), for improving the generalization of molecular property prediction models from theoretical datasets to real-world chemical data. CSIE aims to maximize information gain and reduce redundancy via an iterative EM-like approach using molecular representations augmented with both graph-based and textual features. The work also introduces the AMGNN, an asymmetric molecular interaction graph neural network designed to better capture solute-solvent interactions through directed edge modeling.

**Questions:**

Refer to the weakness.

**Ethical Concerns:**

["NO or VERY MINOR ethics concerns only"]

**Final Justification:**

I am inclined to accept this article. On the one hand, the authors have demonstrated the effectiveness of their method in specific domains (molecular solubility prediction and band gap prediction). On the other hand, data quality is a prevalent issue across all AI4Science fields, making data-centric approaches increasingly important. This article serves as a typical example in this regard. Although it does not explore extensions to other domains, I believe it holds certain reference value.

**Limitations:**

Yes

**Paper Formatting Concerns:**

NA.

**Quality:**

3

**Strengths And Weaknesses:**

## Strength
1. The paper grounds the CSIE approach in submodular function theory and utilizes information-theoretic and geometric measures for informative sample selection, moving beyond simple uncertainty or loss-based criteria.
2. The data-centric approaches hold certain reference value in the field of AI for Science, given that this field typically faces the problem of data confounding.
3. The results show that CSIE enables strong performance even when training on a fraction of the real-world data, and AMGNN outperforms or matches other strong GNN baselines.


## Weakness
1. The use of information gain as defined through average cosine distance in the embedding/HNSW graph is only described in a high-level and is not deeply tied to chemical relevance—there is insufficient empirical analysis or visualization on whether the CSIE-selected core samples are chemically diverse, rare, or cover meaningful classes.
2. I think that if the CSIEFramework is also applied to demonstrate performance improvements across different network models, it would significantly aid in verifying the method's effectiveness.
3. Can the authors clarify the attribution of AMGNN's gains over MMGNN or CGIB? Is it driven solely by the asymmetric message passing or also by the use of multimodal (MolCA) representations?
4. The domain gap issue is important, but the demonstrated improvements are incremental and not conclusively shown to generalize across molecular chemistries or other tasks. I hope to provide some discussion on the generalizability of the method.

---

> ### Author Rebuttal · Authors · 2025-07-29
>
> # Response to Reviewer 24Q7:
>
> Thank you for your thoughtful and constructive feedback. We sincerely appreciate the time and effort you dedicated to reviewing our manuscript. We are encouraged by your recognition of the theoretical grounding of the CSIE framework, its relevance to data-centric AI for Science, and the empirical performance of both CSIE and AMGNN. At the same time, we acknowledge the need for deeper clarification regarding the chemical relevance of information gain, the contributions of different components (e.g., MolCA vs. asymmetric message passing), and the broader generalizability of our approach. Below, we provide a detailed, point-by-point response to each of your insightful comments.
>
> ---
>
> > **W1.** Visual Analytic
>
> Thank you for pointing out the issue. We explain as follows:
>
> - First, in terms of theoretical foundations, the connection between the information gain metric and chemical properties is primarily established through the use of a more effective encoder. In this study, we employ both molecular graph features and semantic features for encoding. Additionally, we utilize **MolCA**, a general-purpose chemical foundation model pre-trained on **PubChem 324k**, to fully extract sample features and ensure the completeness of chemical semantics as much as possible.
>
> - Second, we divide the dataset based on solvent types and evaluate the **sample category distribution before and after CSIE filtering**. It is clearly observable that CSIE effectively alleviates the data redundancy issue, resulting in a **more balanced dataset in terms of class distribution**. Detailed results can be found in **Section 3.3, Figure 3**, along with the corresponding analysis.
>
>
> > **W2.** Application of CISE in different network models
>
> - **CSIE consistently improves model performance across different network architectures**: In **Table 1 on page 7**, the column *CombiSolv-Exp → CombiSolv-Exp* shows the performance of each model trained directly on real experimental data, while *QM-mini → CombiSolv-Exp* represents models pretrained on the CSIE-filtered QM-mini dataset and fine-tuned on the real data. **A comparison of these two columns reveals that most models exhibit significant performance improvements, demonstrating the effectiveness of CSIE**.
>
> - Similarly, **Tables 2 and 3 on page 8** present the performance under different solute/solvent type splits, illustrating **CSIE's enhancement of model generalization ability**. In **Table 9 on page 20 (Appendix)**, we compare the performance gains of CSIE with other core-set selection frameworks across different network models. **The results clearly show that CSIE outperforms the alternatives**.
>
> > **W3.**  Reasons for the performance improvement of AMGNN
>
> - The performance improvement of **AMGNN** over **MMGNN** and **CGIB** is indeed attributed to its **asymmetric message-passing mechanism**. Under both experimental settings—*CombiSolv-Exp → CombiSolv-Exp* and *QM-mini → CombiSolv-Exp*—AMGNN achieves the best performance.
>
> - In the former setting, the model is trained directly on the target dataset without any influence from CSIE, let alone the *MolCA* module within CSIE. In the latter transfer learning scenario, *MolCA* only plays a role during the data filtering process; the filtered *QM-mini* dataset is then fairly used across all models.
>
> - **In summary**, the performance gains of AMGNN are independent of *MolCA*.
>
>
> > **W4.**  Discussion on the generalizability of CSIE and AMGNN
>
> We appreciate the reviewer’s concern regarding the generalizability of our proposed method. While the **dissolution free energy prediction task serves as a focused case study**, both CSIE and AMGNN are designed as **general-purpose tools** for broader applications in molecular machine learning.
>
> **(1) Generalizability of CSIE.**
> CSIE is a **model-agnostic, data-centric framework** aimed at improving the quality and efficiency of theoretical datasets. While it draws inspiration from active learning, it **specifically addresses limitations in uncertainty-based sampling**, which is often unreliable in noisy chemical datasets. To demonstrate its generality, we applied CSIE to a **material band gap prediction task** (see Appendix E.5.1), which is entirely distinct from dissolution. The performance gains observed there confirm that **CSIE is not limited to solubility tasks**, but rather serves as a **universal dataset screening strategy** for scientific machine learning domains where data redundancy and imbalance are common.
>
> **(2) Generalizability of AMGNN.**
>
> AMGNN proposes an **asymmetric molecular graph construction paradigm** that directly encodes the directional interactions between two molecules. Unlike traditional methods that treat molecular pairs **symmetrically**—either through feature concatenation or attention-based fusion (e.g., Uni-Mol, Explainable GNN, CIGIN)—AMGNN captures the fact that **molecular interactions are not order-invariant**.
>
> For example, as stated in line 173, the solvation energy of **ethanol in water** differs significantly from that of **water in ethanol**. This asymmetry is also critical in other molecular relation learning tasks—such as **drug–drug interaction**
>
> By explicitly capturing these asymmetries through **heterogeneous graph modeling**, AMGNN can be **readily applied to other pairwise molecular tasks** such as reaction prediction and toxicity analysis. As discussed in **W2 & W3**, we believe this paradigm opens a promising direction for more **chemically realistic and interpretable molecular interaction learning**.
>
> In summary, while solubility serves as a representative task, the methods we propose—**CSIE for data selection** and **AMGNN for asymmetric modeling**—are both **task-agnostic and widely applicable**, offering general value to the molecular machine learning community.
>
> ---
>
> We greatly appreciate your insightful and constructive comments, which have helped us clarify key aspects of our work and strengthen the manuscript. We hope that our point-by-point responses have addressed your concerns, especially regarding the chemical relevance of information gain, the independent contributions of CSIE and AMGNN, and the generalizability of our approach. If our clarifications and additional analysis have resolved the ambiguities you raised, we would be sincerely grateful if you would consider revising your evaluation. Of course, if you have any further questions or suggestions, we would be more than happy to continue the discussion. Thank you once again for your thoughtful review and valuable feedback—it has been instrumental in improving our work.

---

> ### Author Response · Authors · 2025-08-05
>
> Dear Reviewer,
>
> We sincerely appreciate the time and effort you dedicated to reviewing our manuscript. Your thoughtful and constructive comments significantly contributed to improving the clarity and rigor of our work.
>
> In our recent response, we have carefully addressed your concerns. We would be grateful to know if our replies have addressed your concerns. If there are any remaining issues or areas where you feel further clarification or improvement is needed, we would be more than happy to provide additional details or make further revisions to strengthen the manuscript.
>
> Thank you once again for your valuable feedback and for helping us improve the quality and clarity of our work.
>
> **Best regards,**
>
> *The authors*

---

> ### Author Response · Authors · 2025-08-07
> **We Look Forward to Your Reply**
>
> Dear Reviewer,
>
> We sincerely appreciate the time and effort you have dedicated to reviewing our manuscript. We recently submitted a detailed response addressing all the concerns you raised. **We would like to kindly ask whether our responses have adequately resolved your concerns. If there are any further issues or areas that require additional clarification or improvement, please do not hesitate to let us know. We are more than willing to make further adjustments to ensure the work meets your expectations.**
>
> If our replies have addressed your concerns to your satisfaction, we would be grateful if you could consider reflecting this in your evaluation by providing a positive score.
>
> Thank you once again for your constructive feedback and for helping us improve the quality of our work.
>
> Warm regards,
>
> The authors

---

> > ### Comment · Reviewer_24Q7 · 2025-08-08
> >
> > Thank you for your detailed reply, which has basically addressed my concerns. I apologize for my delayed response.  I will maintain a positive score.

---

> > > ### Author Response · Authors · 2025-08-08
> > >
> > > Dear Reviewer,
> > >
> > > Thanks very much for your insightful and constructive feedback, and we sincerely appreciate the time and effort you have invested in reviewing our paper. We are greatly encouraged by your recognition of our work.
> > >
> > > Thank you again for your careful guidance. These feedbacks have played a crucial role in improving our research work.
> > >
> > > Best regards,
> > >
> > > The authors

---

### Comment · Area_Chair_nDyD · 2025-08-05

Dear Reviewers:

The Author-Reviewer Discussion Period will remain open until August 8 (AoE).

Your active participation during this phase is essential. Please:

- Read the author responses and other reviews.

- Engage constructively with authors to clarify any concerns.

Thanks to those who have already begun their discussions, and to all of you for your hard work with these reviews.

AC

---

### Note · Authors · 2025-08-13

**Dear (Senior) ACs and Reviewers,**

We would like to express our heartfelt thanks for the time, care, and expertise you have devoted to reviewing our manuscript. We are truly grateful for the constructive feedback and thoughtful suggestions, which have been invaluable in helping us refine and strengthen our work.

**Molecular chemistry tasks in AI4S often face domain gap issues from theoretical data to actual data. This problem is often affected by two aspects: data distribution bias and insufficient model generalization.** Therefore, our manuscript proposes a universal Core Subset Iterative Selection (CSIE) framework and an Asymmetric Graph Neural Network (AMGNN) with strong generalization ability for molecular relationship learning. We have theoretically demonstrated the effectiveness of CSIE and conducted strong empirical research and visualization analysis on molecular dissolution and bandgap prediction. We have also demonstrated the effectiveness and generalization of AMGNN on multiple datasets and experimental settings. CSIE and AMGNN complement each other, and we hope that both can make positive contributions to the development of the AI4S community.

During the rebuttal process, the reviewers provided constructive feedback on some of the manuscript's shortcomings, with some commonalities and major concerns including:
1. **The correlation and respective contributions between CSIE and AMGNN**
2. **The impact of dataset bias on CSIE**
3. **Discussion on the generalization of methods**

We are honored that we have almost addressed all the concerns of the reviewers. We actively discussed the connections between components and analyzed the theoretical basis for the generalizability of our methods. Besides, we supplemented additional experiments by modifying the dataset to demonstrate the effectiveness and robustness of CSIE and AMGNN. Two reviewers (**24Q7, vRGh**) expressed positive recognition of our work and rebuttal content, while the other two reviewers (**2Vf7, Lzfx**) stated that our rebuttal addressed concerns and were willing to increase the score. We have benefited greatly from the discussion among the reviewers regarding the writing and research content of the article. We promise to cooperate with the reviewers to revise the article and add additional experiments to the camera ready version.

Once again, we sincerely appreciate your thoughtful evaluation, the generous feedback, and the opportunity to clarify and improve our work.

---

### Decision · Program_Chairs · 2025-09-17

**Decision:**

Accept (poster)

**Comment:**

This paper proposes Core Subset Iterative Extraction (CSIE), a sample selection framework for improving molecular property prediction, and Asymmetric Molecular Graph Neural Network (AMGNN), a GNN architecture for solute-solvent interactions. The idea is considered theoretically grounded (24Q7, 2Vf7, vRGH), impactful for AI4Science (24Q7, vRGh), and novel (Reviewers 2Vf7, vRGh). Although some weaknesses remain, including limited clarity, modest performance gains (2Vf7, Lzfx), fortunately, the authors have addressed the main issues through a detailed rebuttal providing additional clarification and experiment results.